# TRACKING THE FEATURE DYNAMICS IN LLM TRAINING: A MECHANISTIC STUDY

## ABSTRACT

Understanding training dynamics and feature evolution is crucial for the mechanistic interpretability of large language models (LLMs). Although sparse autoencoders (SAEs) have been used to identify features within LLMs, a clear picture of how these features evolve during training remains elusive. In this study, we (1) introduce SAE-Track, a novel method for efficiently obtaining a continual series of SAEs, providing the foundation for a mechanistic study that covers (2) the semantic evolution of features, (3) the underlying processes of feature formation, and (4) the directional drift of feature vectors. Our work provides new insights into the dynamics of features in LLMs, enhancing our understanding of training mechanisms and feature evolution.

## 1 INTRODUCTION

As LLMs increase in size and complexity, understanding their internal mechanisms has become a critical challenge. *Polysemanticity* – where single neurons respond to mixtures of unrelated inputs – complicates interpretability. Sparse Autoencoders (SAEs) offer a promising approach to this challenge by disentangling overlapping features, enabling the extraction of more interpretable structures from activations (Bricken et al., 2023; Cunningham et al., 2023). Recent work demonstrates that SAEs can scale to models like Claude 3 Sonnet (Templeton et al., 2024), uncover geometric structures in representations (Li et al., 2024; Engels et al., 2024), and even intervene in model behavior (Chalnev et al., 2024; Yin et al., 2024; Farrell et al., 2024).

Researchers have also compared features across different settings, such as different layers (Jack Lindsey et al., 2024; Balagansky et al., 2024), between base and fine-tuned models (Jack Lindsey et al., 2024; Connor Kissane et al., 2024; Taras Kutsyk, 2024), and across different model scales (Lan et al., 2024). However, LLMs' training dynamics remain underexplored, despite their importance in understanding the mechanisms underpinning LLM behavior.

While prior work in mechanistic interpretability has shed light on individual phenomena related to training dynamics – for instance, Olsson et al. (2022) on induction head formation and in-context learning, Nanda et al. (2023) on grokking phenomenon of generalization, Qian et al. (2024) on trustworthiness emergence in LLM pre-training, Ren et al. (2024) on interactions, and Li et al. (2023) on topic-structure development – these studies often remain narrowly scoped, focusing on isolated abilities or providing static snapshots of network behavior. What is largely missing, therefore, is a unified and continuous framework that *systematically monitors how fundamental feature representations evolve – both semantically and geometrically – throughout the entire training trajectory*.

To address this critical gap, this paper introduces **SAE-Track** to effectively track feature evolution and conduct a **mechanistic study** based on this framework, covering the semantic evolution of features, feature formation, and feature vector drift analysis.

Our main contributions are as follows:

- **SAE-Track**: A novel method for efficiently obtaining a continual series of SAEs, providing the foundation for a detailed mechanistic study of feature evolution in LLMs (section 2).
- **Semantic Evolution Phases and Patterns**: Identification of three characteristic phases – *Initialization & Warmup*, *Emergent*, and *Convergent* – and three primary transformation patterns:

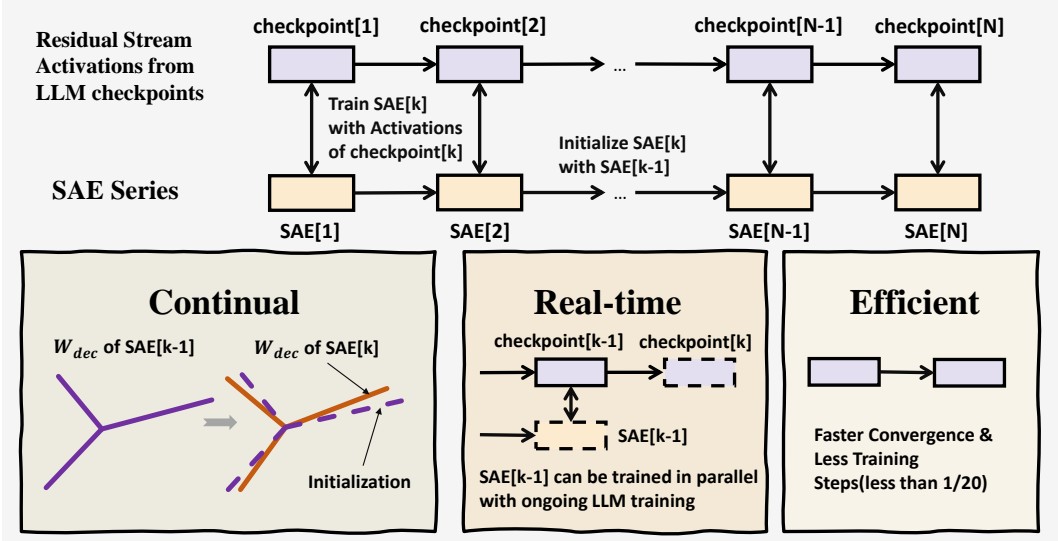

Figure 1: **SAE-Track Framework.** Sparse Autoencoders (SAEs) are trained on residual stream activations from sequential LLM checkpoints. Each SAE[k] is initialized from the previous SAE[k-1], enabling **continual** tracking, **real-time** parallel training, and **efficient** computation with reduced steps (e.g., <1/20) for subsequent SAEs.

*Maintaining*, *Shifting*, and *Grouping*, capturing the gradual transition from a randomized transformer to a well-trained one (section 3).

- **Feature Formation**: Geometric modeling of feature formation as the convergence of datapoints into localized regions, with a novel Progress Measure that captures both the gradual nature of this process and the distinct dynamics of token-level and concept-level features (section 4).
- **Feature Drift and Trajectory Analysis**: Analysis of feature drift, revealing that feature directions undergo significant, three-phase adjustments. Counter-intuitively, this drift persists even after features are semantically "formed", with full stabilization only occurring late in training (section 5).
- **Extensibility Validation**: Extensive experiments with open-source LLMs (Pythia and Stanford CRFM GPT-2) of varying scales (124M to 1.4B) and across different residual stream layers, confirming the extensibility and generality of our approach. (Pythia-410M-deduped, layer 4 for main paper results; additional models and layers in Appendix K.) We have tested our method on as many feasible models as possible within our computational constraints and available checkpoints (See limitations discussed in Appendix A.)

## 2 SAE-TRACK: GETTING A CONTINUAL SERIES OF SAEs

### 2.1 PRELIMINARIES: SPARSE AUTOENCODERS (SAEs)

We employ Sparse Autoencoders (SAEs) (Bricken et al., 2023; Templeton et al., 2024), an unsupervised method to learn sparse, interpretable features from data such as LLM activations $\mathbf{x} \in \mathbb{R}^D$. An SAE is trained to reconstruct $\mathbf{x}$ as $\widehat{\mathbf{x}}$ from a parsimonious set of learned dictionary features weighted by their activations $f_i(\mathbf{x})$. This is achieved by minimizing the loss function $\mathcal{L}$ which combines reconstruction error with an L1 sparsity penalty on $f_i(\mathbf{x})$:

$$\mathcal{L} = \mathbb{E}_{\mathbf{x}} \left[ \|\mathbf{x} - \widehat{\mathbf{x}}\|_2^2 + \lambda \mathcal{L}_1 \right]. \tag{1}$$

The detailed mathematical formulation of the SAE architecture, including the computation of feature activations $f_i(\mathbf{x})$, the reconstruction $\widehat{\mathbf{x}}$, and specific forms of the L1 penalty $\mathcal{L}_1$, is provided in Appendix B.

## 2.2 KEY INTUITIONS

SAEs are trained on activations that evolve incrementally during gradient descent. Leveraging this continuity enables efficient feature extraction and reduces noise when analyzing feature evolution across checkpoints.

Formally, let $F^{(l,t)}$ denote the transformer model at layer $l$ and step $t$, parameterized by $\Theta^{(<l,t)}$. The activation for token $q$ in context $\mathcal{C}$ is:

$$\mathbf{x}^{(l,\mathcal{C},q,t)} = F^{(l,t)}(\mathcal{C}, q; \Theta^{(<l,t)}). \tag{2}$$

For simplicity, we might omit some of $(l, \mathcal{C}, q, t)$ in subsequent discussions.

**Theorem 2.1** (**Training-Step Continuity**). *Assume (1) the gradient norm $\|\nabla_{\Theta^{(<l)}}\mathcal{L}(\Theta)\| \leq G$ is bounded, and (2) the function $F^{(l)}$ is Lipschitz continuous with respect to $\Theta^{(<l)}$, with constant $L$. Let $\epsilon$ be a small positive constant. If the learning rate satisfies $\eta < \frac{\epsilon}{LG}$, then the activations evolve incrementally:*

$$\left\|\mathbf{x}^{(l,t)} - \mathbf{x}^{(l,t-1)}\right\| < \epsilon, \tag{3}$$

The proof is provided in Appendix C. This theorem formalizes the intuition that, under bounded gradients and a suitably small learning rate, each parameter update induces only a minor shift in layer activations. Although real-world training may deviate from this idealized assumption, activation continuity motivates our approach to efficiently track feature evolution using a continual series of SAEs.

## 2.3 METHODOLOGY: RECURRENT INITIALIZATION

To efficiently track feature evolution, we propose **SAE-Track**, which constructs a continual series of SAEs via **recurrent initialization**: each SAE[$k$] is initialized from SAE[$k-1$] (with SAE[1] randomly initialized) and trained on activations from checkpoint[$k$]. As illustrated in figure 1, SAE-Track is designed to be **continual**, **real-time**, and **efficient**. It maintains feature continuity by leveraging recurrent initialization, supports real-time training alongside LLM checkpoints, and significantly reduces training cost by requiring fewer training steps for subsequent SAEs.

To validate the effectiveness of this approach, we conducted a series of comparative experiments (see Appendix I.1 and Appendix I.2). Appendix I.1 demonstrates that SAE-Track not only significantly accelerates convergence but also produces individual SAEs that exhibit similar behavior to those trained using conventional methods, ensuring that the learned representations remain consistent with standard training approaches. Appendix I.2 further confirms that the observed feature convergence is not merely an artifact of the SAE-Track process, but a fundamental property of the model's feature dynamics.

SAE-Track provides an efficient and systematic approach to analyze feature dynamics throughout the training process. Based on the SAE series generated by SAE-Track, we can study the semantic evolution of features (section 3), examine their formation from a geometric perspective (section 4), and analyze how features undergo directional changes (section 5).

## 3 SEMANTIC EVOLUTION OF FEATURES

This section delves into the semantic evolution of features as LLMs undergo training, leveraging the series of SAEs generated by SAE-Track. Our analysis method involves **tracking the interpretation of fixed feature indices (IDs) across these sequential SAEs**, which lead to our primary conclusion regarding the phases and patterns of semantic evolution:

---

**Conclusion 1**

Based on the taxonomy distinguishing between **token-level** and **concept-level features**—itself an observation from their differing activation behaviors (see Definition 3.1 and Definition 3.2)—we identify a characteristic **three-phase process** governing their semantic evolution throughout training. Furthermore, this distinction allows us to categorize the observed feature transformation patterns, as exemplified in figure 2, into **three primary patterns**.

---

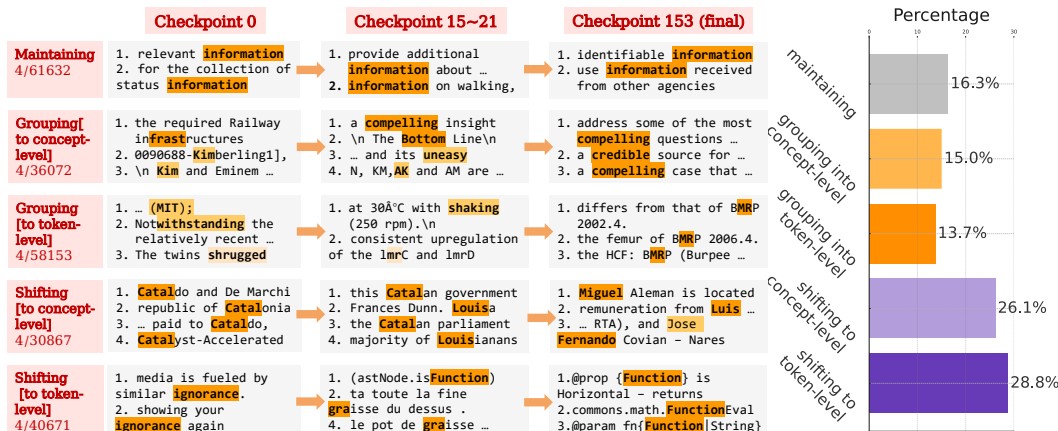

Figure 2: **Feature transition patterns.(on Pythia-410m.)** The **left panel** displays examples of feature transitions—maintaining, grouping (into concept/token-level feature), and shifting (to concept/token-level feature)—across training checkpoints. The **right panel** shows the statistics of each transition type (on 256 random sampled features).

The distinction between token-level and concept-level features is foundational to understanding their evolution. As evidenced by their differing activation patterns over the training examples shown in figure 2 (Left), we define them as follows:

**Definition 3.1** (**Token-Level Feature**). *A **token-level feature** predominantly activates for a specific token, such as "century."*

**Definition 3.2** (**Concept-Level Feature**). *A **concept-level feature** activates across a set of tokens that are semantically related to a broader concept. For instance, "authentication" and "getRole()" may activate for the concept "user authentication." This category also encompasses weak concept-level features, such as those based on morphological variants("arrive" and "arrives"), as detailed in Appendix F.*

Token-level features are typically present from initial checkpoints, while concept-level features emerge more gradually. This evolution, observable by comparing early versus late checkpoint examples in figure 2 (Left), can be described in three stages:

- **Initialization and Warmup.** Token-level features are present from the outset, while other activations often appear as noise with limited semantic association beyond individual tokens, as seen in early checkpoint examples (e.g., "ckpt 0" in figure 2).
- **Emergent Phase.** Concept-level features begin to form, with activations grouping around semantically related tokens, while token-level features still exist and noise features diminishes, visible in intermediate checkpoints (e.g., "ckpt 15-21" in figure 2).
- **Convergent Phase.** Both feature types stabilize into interpretable states, with concept-level features forming coherent semantic groups by later checkpoints (e.g., "ckpt 153(final)" in figure 2).

Further examination of individual feature ID evolutions, by analyzing changes in their top-k activating examples as illustrated in figure 2 (Left), reveals three distinct transformation patterns. Their statistics, from a sample of 256 features, is shown in figure 2 (Right). These patterns are:

- **Maintaining.** Token-level features often persist across checkpoints with consistent token activations, as exemplified by the first row of examples in figure 2 (Left).
- **Shifting.** Some features alter their primary semantic association during training. These shifts can be towards a new **token-level** representation (exemplified in the fifth row of figure 2 (Left)) or an evolution into a broader **concept-level** one (exemplified in the fourth row of figure 2 (Left)), reflecting representational reorganization (see Appendix H).
- **Grouping.** Features initially exhibiting noisy or diffuse activations gradually organize into semantically coherent structures. This can result in the formation of new **concept-level** features (exemplified in the second row of figure 2 (Left)) or new **token-level** features (exemplified in the third row of figure 2 (Left)).

**Our proposed transformation patterns form a comprehensive taxonomy**, since they encompass all observed semantic developmental pathways from initial (token-level or noise) to mature (token-level or concept-level) feature states.

While this qualitative characterization offers valuable insights, its interpretative nature (common in SAE analysis though) has inherent limitations. **To go beyond these limitations and provide more robust validation, we conduct further quantitative mechanistic investigations in subsequent sections.** Both Section 4 (on feature formation) and Section 5 (on feature drift) quantitatively support the three-phase development model, with the latter also providing a geometric complement to the semantic analysis herein.

## 4 ANALYSIS OF FEATURE FORMATION

This section analyzes feature formation, from noisy activations to meaningful representations. We first frame this geometrically as datapoint convergence into local regions. We then use a novel Progress Measure to quantitatively show this process is gradual, and to explain the distinct emergence and learning dynamics of token-level versus concept-level features.

> **Conclusion 2: Feature Formation Mechanisms**
>
> Feature formation involves **geometric convergence of activations into local regions**. Our Progress Measure confirms this is **gradual**, elucidating **token-level feature initial existence versus later concept-level feature learning**, and supports a three-phase development model.

### 4.1 GEOMETRIC PERSPECTIVE AND QUALITATIVE ILLUSTRATION OF FEATURE FORMATION

Existing studies often emphasize the final state of training or assume that features are inherently monosemantic (Bricken et al., 2023; Templeton et al., 2024; Elhage et al., 2022). However, training a SAE at any model checkpoint reveals features that define separable regions in the activation space. While the specific properties of these features may vary across checkpoints, they can all be understood from a unified geometric perspective: they represent distinct regions within the activation space.

From this perspective, the role of the SAE is to identify and isolate these regions. Formally, the encoder for feature $i$ can be expressed as:

$$f_i(\mathbf{x}) = \text{ReLU}\left(\widehat{\mathbf{W}}_i \cdot \mathbf{x} + \widehat{b}_i\right), \tag{4}$$

where $\widehat{\mathbf{W}}_i = \mathbf{W}_{i,:}^{\text{enc}}$ and $\widehat{b}_i = b_i^{\text{enc}} - c \cdot \mathbf{W}_{i,:}^{\text{enc}} \cdot \mathbf{b}^{\text{dec}}$. Using this, we can define a feature region based on activation strength.

The function $f_i(\mathbf{x})$ naturally divides the activation space into regions via the ReLU function, where $\widehat{\mathbf{W}}_i \cdot \mathbf{x} + \widehat{b}_i > 0$ defines a region for $\mathbf{x}$. However, this division ignores the impact of activation strength, which is critical for semantic fidelity. As noted by Bricken et al. (2023), higher activation levels often indicate stronger associations with specific tokens or concepts. We formally define such a region as follows:

**Definition 4.1** (**Feature Region by Activation Strength**). *A region corresponding to feature $i$ with activation strengths in the range $[L, U)$ is defined as:*

$$\mathcal{R}_i^{[L,U)} = \{\mathbf{x} \mid L \leq \left(\widehat{\mathbf{W}}_i \cdot \mathbf{x} + \widehat{b}_i\right) < U\}. \tag{5}$$

figure 3 (Top Left) provides a 2D toy example showing how varying activation levels, i.e., different values for $L$ in Definition 4.1, can define more precise regions. To focus our analysis on semantically meaningful feature activations, we typically consider datapoints that elicit high activation strengths for a given feature $i$. In practice, this is often achieved by identifying the set of input contexts and tokens $(\mathcal{C}, q)$ that produce the top $k$ highest activation values for feature $i$ (e.g., $k = 25$ in our visualizations and analyses). This set of top-$k$ activating datapoints effectively defines the most salient region of activity for feature $i$. figure 3 (Bottom Left) conceptually illustrates how such distinct high-activation regions for different features can partition the activation space, highlighting the SAE's objective to identify these separable, semantically rich areas.

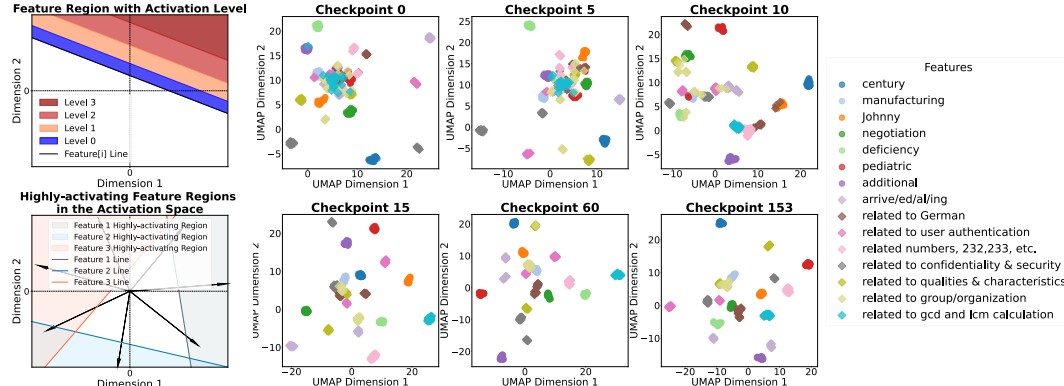

Figure 3: **Feature Formation: Geometric Concepts and UMAP Visualization. Top Left**: Feature regions defined by varying activation levels. **Bottom Left**: Distinct feature regions derived from high activations, highlighting the partitioning within the activation space. **Right**: UMAP visualizations of activation set evolution for various features (Pythia-410m-deduped), from initial randomness (Checkpoint 0) to coherent clusters (Checkpoint 153), distinguishing token-level (circular) and concept-level (diamond) features.

Feature formation, therefore, describes how datapoints with similar semantics converge into these high-activation regions associated with specific features. To study this phenomenon, we first identify at the final checkpoint ($T_{\text{final}}$) the set $\mathcal{D}_i$ comprising the (context, token) pairs $(\mathcal{C}, q)$ that yield the top $k$ activations for each feature $i$. The activation set for these specific $(\mathcal{C}, q)$ pairs at any given training step $t$ is then defined as:

$$\mathcal{A}_i^t = \{F^{(t)}(\mathcal{C}, q; \Theta^{(t)}) \mid (\mathcal{C}, q) \in \mathcal{D}_i\}. \tag{6}$$

Here, $\mathcal{A}_i^t$ represents the activations at checkpoint $t$ for this consistently defined set of $\mathcal{D}_i$ datapoints. Thus, studying feature formation involves analyzing the dynamics of $\{\mathcal{A}_i^t\}_{i=0}^{F-1}$ across training.

To visually inspect the convergence of activation sets $\{\mathcal{A}_i^t\}_{i=0}^{F-1}$, **we use UMAP projections** (McInnes et al., 2020) (see figure 3, Right). Specifically, these plots qualitatively depict the geometric convergence of feature activations, where initially dispersed concept-level features (diamonds) progressively form cohesive clusters throughout training, contrasting with token-level features (circles) that often exhibit some degree of clustering from early stages (e.g., Checkpoint 0). While offering valuable intuition, these UMAP plots serve primarily as a qualitative illustration, motivating **the rigorous quantitative analysis of feature formation presented next.**

## 4.2 Quantitative Analysis with a Progress Measure

To address the question of whether feature formation is a phase transition or a progressive process, and to provide a more objective assessment than visual inspection (figure 3(b)), we propose the **Feature Formation Progress Measure**. This metric quantifies the degree to which a feature becomes well-formed during training by comparing the similarity within its representative datapoints (identified via top-$k$ activations at $T_{\text{final}}$, as per section 4.1) to a baseline derived from randomly sampled, unrelated datapoints.

**Definition 4.2 (Feature Formation Progress Measure).** *The metric $M_i(t)$ at training step $t$ is defined as:*

$$M_i(t) = \overline{Sim}_{\mathcal{A}_i^t} - \overline{Sim}_{\mathcal{A}_{random}}, \tag{7}$$

*where $\mathcal{A}_i^t$ is the activation set of the top-$k$ datapoints for feature $i$ at step $t$, $\mathcal{A}_{random}$ represents a set of randomly sampled datapoints, and:*

$$\overline{Sim}_{\mathcal{A}_i^t} = \frac{2}{|\mathcal{A}_i^t|(|\mathcal{A}_i^t|-1)} \sum_{\substack{x_k, x_j \in \mathcal{A}_i^t \\ j < k}} Sim(x_k, x_j), \tag{8}$$

$$\overline{Sim}_{\mathcal{A}_{random}} = \frac{2}{|\mathcal{A}_{random}|(|\mathcal{A}_{random}|-1)} \sum_{\substack{x_k, x_j \in \mathcal{A}_{random} \\ j < k}} Sim(x_k, x_j). \tag{9}$$

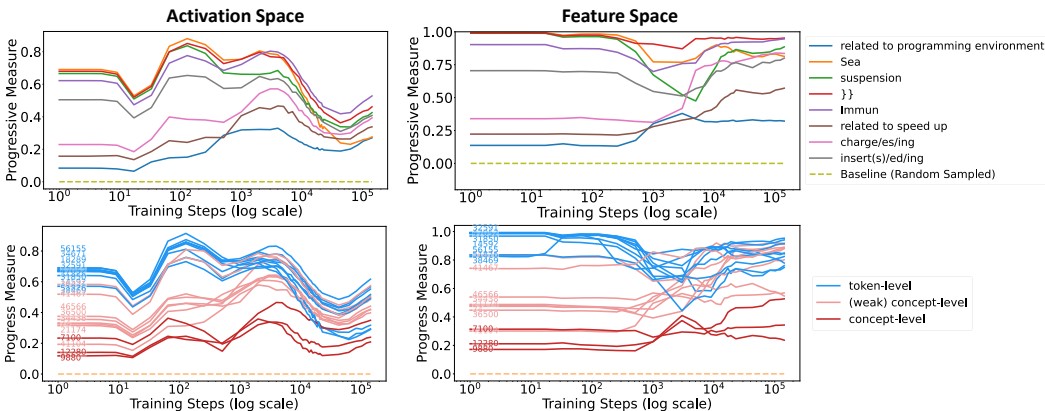

Figure 4: **Feature Dynamics Across Training Steps.** Progress measures for the **activation space** (Left) and **feature space** (Right) of **Pythia-410m-deduped** across training steps (log scale). The top row shows the dynamics of features with manual annotation as examples, including concept-level and token-level ones. The bottom row presents a broader set of randomly sampled features, categorized into **token-level** (blue), **(weak) concept-level** (light red), and **concept-level** (dark red).

*We also define a corresponding measure $M_i^{feature}(t)$ in feature space using the SAE-encoded features $\mathcal{F}_i^t$ and $\mathcal{F}_{random}$, providing a finer-grained perspective (see definition in section D).*

The Progress Measure, applied using cosine similarity, offers quantitative insights into the mechanisms of feature formation. **figure 4 quantitatively demonstrates these dynamics in both activation and feature spaces, revealing several key characteristics of this process.** Specifically:

- Most features undergo **predominantly a gradual formation process**, rather than an abrupt transition, evidenced by the smooth evolution of their Progress Measure curves in figure 4.
- figure 4 verifies the **initial existence of token-level features** and the **progressive learning of abstract concept-level features** (mentioned in section 3). As shown in figure 4 (bottom row), token-level features (blue curves) typically exhibit high value from the start, contrasting with concept-level features (dark red curves) which generally start lower and increase gradually throughout training.
- Concept features follow a **three-phase development process** (Initialization/Warmup, Emergent, Convergent at section 3). This is quantitatively supported by the characteristic three-phase 'low-rise-stable' trajectory of their Progress Measure curves in figure 4 (e.g., bottom-row dark red curves and several top-row examples), with the plateau often being more pronounced in the feature space plots (right column).
- A potential spectrum exists between purely token-level and abstract concept-level features, where weak concept-level features (light red curves) like morphological features (top-row examples) often display patterns intermediate to the distinct token-level (blue) and concept-level (dark red) ones (further discussed in Appendix F).

## 5 ANALYSIS OF FEATURE DRIFT

It remains elusive whether feature directions (i.e., crucial geometric components identified by SAEs) undergo significant drift throughout training or become fixed early on. Understanding this dynamic is vital for a comprehensive picture of how feature representations stabilize. Accordingly, this section investigates the evolution of these feature directions, leading to two important conclusions about their dynamics during the training process.

### 5.1 DECODER VECTOR EVOLUTION

We examine the evolution of feature directions by analyzing their normalized decoder vectors $\frac{\mathbf{W}_{:,i}^{dec}}{\|\mathbf{W}_{:,i}^{dec}\|}$ (following Templeton et al. (2024)) and calculating their cosine similarity with their respective final directions at $t = \text{final}$.

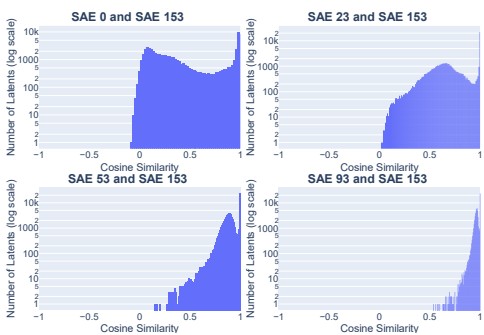 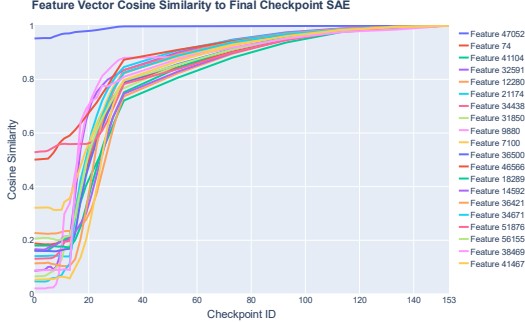

(a) Distribution of cosine similarity between decoder vectors at intermediate training checkpoints and the final checkpoint.

(b) Cosine similarity progression for sampled features to their final direction across training checkpoints. Each colored line represents an individual feature.

Figure 5: **Decoder Vector Evolution.** (a) Distributions show the global trend of feature direction alignment over training. (b) Individual feature trajectories illustrate the phased nature of this alignment. All analyses conducted on **Pythia-410m-deduped**.

---

**Conclusion 3.1: Decoder Vector Dynamics**

Our analysis of decoder vector evolution reveals that most feature directions undergo **significant drift** throughout training, challenging assumptions of early stability. This alignment to their final states is also a **three distinct phases** that similar to the features' semantic evolution.

---

The evidence supporting these conclusions is presented in figure 5. The widespread and significant nature of feature drift is demonstrated by figure 5(a). These distributions of cosine similarities between current and final feature directions, initially broad and skewed towards lower values, markedly shift towards 1 as training progresses, indicating a global, gradual alignment process.

The three-phase pattern of this alignment is evident in figure 5(b), which displays the cosine similarity progression for a collection of sampled features. These trajectories collectively illustrate: (1) an *Initialization & Warmup* phase, often with low or stable similarity; (2) an *Emergent* phase, marked by a rapid increase in similarity as features orient towards their final directions; and (3) a *Convergent* phase, where alignment plateaus at high values. As noted, these dynamic phases in directional convergence similar to the semantic evolution stages discussed in Section 3.

### 5.2 TRAJECTORY ANALYSIS

To gain a more granular understanding of how individual feature directions evolve continuously, we build upon the preceding analysis by examining their full trajectories across training checkpoints. This requires defining both the trajectory itself and the concept of a "formed" feature.

**Definition 5.1** (**Feature Trajectory**). *Let $\mathbf{W}^{dec}_{:,i}[t]$ denote the decoder vector for feature $i$ at training checkpoint $t$, where $t \in \{1, \ldots, T_{final}\}$. The trajectory of feature $i$ is defined as the sequence of its decoder vectors:*

$$\mathcal{J}_i = \left\{ \mathbf{W}^{dec}_{:,i}[1], \mathbf{W}^{dec}_{:,i}[2], \ldots, \mathbf{W}^{dec}_{:,i}[T_{final}] \right\}. \tag{10}$$

**Definition 5.2** (**Formed Feature**). *A feature is considered formed once it acquires and maintains a stable semantic meaning. Features exhibiting "maintaining" behavior (Section 3) are considered formed throughout their observed duration. Features undergoing "shifting" or "grouping" are deemed formed from the checkpoint at which they attain a stable semantic role that persists until $T_{final}$.*

Our analysis of these feature trajectories (Definition 5.1) reveals a counter-intuitive yet crucial insight into their stabilization dynamics, as summarized in Conclusion 4. Specifically, we find that the semantic stabilization of a feature does not necessarily coincide with the stabilization of its geometric direction.

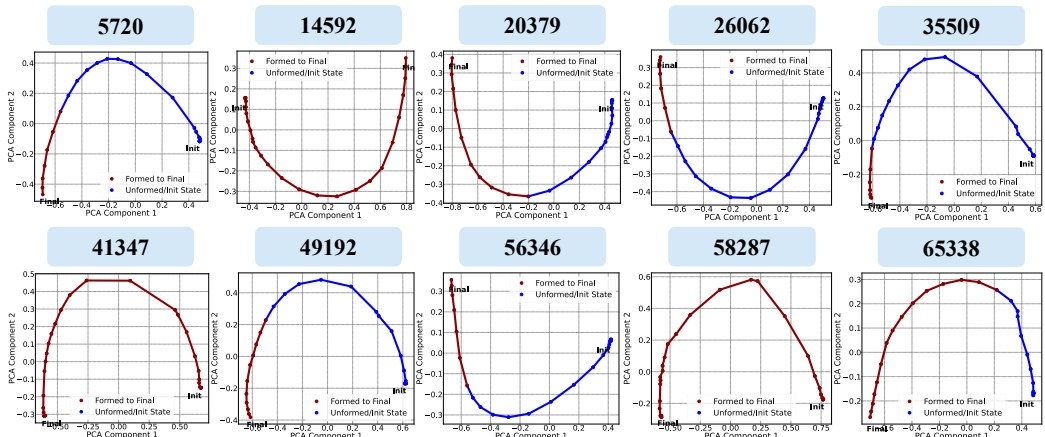

Figure 6: **Feature Trajectories.** Trajectories of (random sampled) decoder vectors represent the directional change of features across training checkpoints. "Dark red" indicates features that are considered "formed," i.e., they have gained semantic meaning and generally remain stable semantic meaning until the final state. "Blue" indicates features that are still unformed or in the initial stage. Conducted on **Pythia-410m-deduped**.

> **Conclusion 3.2: Drift of Semantically Stable Features**
>
> Trajectory analysis reveals a counter-intuitive dynamic: **individual feature directions often continue to drift and geometrically reorganize even after their semantic meaning has stabilized** (i.e., they are "formed"). This underscores that while a feature's semantic role may crystallize, its precise directional embedding undergoes further refinement until the overall feature ensemble converges.

This continued geometric drift of semantically "formed" (Definition 5.2) features is vividly illustrated in figure 6: "formed" segments (dark red) still exhibit noticeable movement. This demonstrates that an individual feature's direction can evolve even after its semantic identity has stabilized. It is indeed counter-intuitive that a single feature can maintain such clear semantic constancy while its geometric direction is still undergoing change, highlighting a distinct decoupling of semantic and directional stabilization at the individual feature level. This persistent directional adjustment of individual features reflects ongoing changing within the overall feature space. Such refinement of individual components is not contradictory to the system's broader convergent phase; rather, it describes how individual features continue to optimize their geometric placement as the entire network settles towards a global equilibrium.

## 6 CONCLUSION

In this paper, we conduct a comprehensive mechanistic analysis of feature evolution in LLMs during training: (1) We propose SAE-Track, a novel method for efficiently obtaining a continual series of SAEs across training checkpoints, providing the foundation for detailed mechanistic studies. (2) We systematically characterize semantic evolution by identifying comprehensive patterns and phases. (3) We mechanistically investigate feature formation, modeling it as the geometric convergence of datapoints into localized regions. Our proposed progress measure captures this gradual process while effectively distinguishing the differing dynamics of token-level and concept-level features. (4) We analyze feature drift, finding directions undergo significant, three-phase adjustments. Counter-intuitively, this drift persists even after features are semantically formed, with full stabilization only occurring late in training. Our work provides a detailed understanding of how features evolve throughout training, demystifying training dynamics from the perspective of SAE-based analysis.

ETHICS STATEMENT

This work adheres to the ICLR Code of Ethics. Our study focuses on mechanistic interpretability of large language models through analysis of publicly available models and datasets, without involving human subjects, private data, or sensitive applications. We therefore do not foresee any direct ethical concerns. The framework and findings are intended solely for scientific understanding and contribute to the development of more transparent and trustworthy AI systems.

REPRODUCIBILITY STATEMENT

We have made every effort to ensure the reproducibility of both our framework and our empirical study. The design of SAE-Track, including its recurrent initialization and theoretical formulation, is presented in Section 2, while the experimental setup—covering model families, checkpoint schedules, etc.—is detailed in Appendix I. An anonymized repository containing the full implementation and experiment scripts is available at https://anonymous.4open.science/r/SAE-Track-ICLR26.

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

# APPENDIX

## A   LIMITATIONS

**Model Architecture and Scale.**    Our analysis is currently restricted to mid-scale autoregressive language models that offer complete and publicly available training checkpoints. Specifically, our work utilizes the **Pythia** suite (Biderman et al., 2023) and the **Stanford-CRFM GPT-2 small/medium** models (Karamcheti et al., 2021), both of which are natively supported by the `TransformerLens` library (Nanda & Bloom, 2022).

This focus is driven by significant constraints. Analyzing considerably larger models is computationally prohibitive due to the resource demands of processing numerous intermediate checkpoints. More critically, most contemporary state-of-the-art open source models lack comprehensive public releases of these intermediate training records. As detailed in Table 1, while some models, particularly those developed for scientific research, offer extensive checkpoints, a majority of prominent SOTA architectures provide limited or no such access. Current open-source practices often prioritize releasing final model weights but not the intermediate checkpoints vital for in-depth studies of pre-training dynamics and feature evolution.

Table 1: **Public availability of intermediate training checkpoints.** Only the first block (above the mid-rule) releases intermediate checkpoints. **TransformerLens** indicates native compatibility with the TRANSFORMERLENS interpretability library. * Only include intermediary checkpoints. [†] Snapshots are only shared with selected researchers upon request; no public dump exists.

| Model family | Arch. / Lic. | Size | # ckpts | TransformerLens |
|---|---|---|---|---|
| Pythia (deduped) (Biderman et al., 2023) | GPT-NeoX / Apache-2.0 | 70M–12B | 154 | ✓ |
| CRFM GPT-2 (Karamcheti et al., 2021) | GPT-2 / Apache-2.0 | 124M / 355M | 609 | ✓ |
| Amber-7B (Liu et al., 2023) | LLaMA-7B / Apache-2.0 | 7B | 360 | ✗ |
| Crystal-7B (Liu et al., 2023) | LLaMA-7B / Apache-2.0 | 7B | 143 | ✗ |
| BLOOM 176B (Scao et al., 2023) | Transformer / RAIL | 176B | 21* | ✗ |
| OLMo (Groeneveld et al., 2024) | Transformer / Apache-2.0 | 1B / 7B | ∼1454 | ✗ |
| OLMo2 (OLMo et al., 2025) | Transformer / Apache-2.0 | 7B / 13B | ∼964 | ✗ |
| OPT 125M–175B (Zhang et al., 2022) | Transformer / NC | 125M–175B | request[†] | ✓(no intermediate) |
| LLaMA-3 (Grattafiori et al., 2024) | Transformer / Llama 3 | 8B–405B | ✗ | ✗ |
| Gemma 2 (Team et al., 2024) | Transformer / Gemma | 2–27B | ✗ | ✗ |
| Qwen 3 (Yang et al., 2025) | Transformer / Apache-2.0 | 0.6B–32B | ✗ | ✗ |

Therefore, considering the limited availability of comprehensive SOTA checkpoints, the necessity of `TransformerLens` compatibility, and our computational capacity, the Pythia and Stanford-CRFM GPT-2 families were selected as the most suitable for this study.

**Human Annotation and Semantic Interpretation.**    In our study, the semantic labeling of features extracted by sparse autoencoders (SAEs) depends on the manual inspection of top-$k$ activating contexts, subsequently followed by the assignment of human-interpretable descriptors. Although this annotation method is standard within SAE literature(Bricken et al., 2023; Templeton et al., 2024), it inherently introduces subjectivity and restricts scalability. Developing automated interpretation pipelines with LLM represents an important avenue for future research to address these limitations.

## B  DETAILED SAE FORMULATION

This appendix provides the detailed mathematical formulation of the Sparse Autoencoders (SAEs) used in our work, following (Bricken et al., 2023; Templeton et al., 2024).

Let $\mathbf{x} \in \mathbb{R}^D$ denote the input activations (e.g., from an LLM's residual stream), where $D$ is the dimensionality of the activation space. The SAE learns a dictionary of $F$ features, and its operations can be described as follows:

**Encoder.**  Feature activations $f_i(\mathbf{x})$ for each feature $i$ are computed by the encoder:

$$f_i(\mathbf{x}) = \text{ReLU}\left(\mathbf{W}^{\text{enc}}_{i,:} \cdot (\mathbf{x} - c \cdot \mathbf{b}^{\text{dec}}) + b^{\text{enc}}_i\right). \tag{11}$$

Here, $\mathbf{W}^{\text{enc}} \in \mathbb{R}^{F \times D}$ are the encoder weights, $\mathbf{b}^{\text{enc}} \in \mathbb{R}^F$ are the encoder biases. The term $c \cdot \mathbf{b}^{\text{dec}}$ involves the decoder bias $\mathbf{b}^{\text{dec}} \in \mathbb{R}^D$ and a constant $c \in \{0, 1\}$ that determines if the decoder bias is subtracted from the input before encoding.

**Decoder.**  The SAE reconstructs the input activations as $\widehat{\mathbf{x}}$ using the decoder:

$$\widehat{\mathbf{x}} = \mathbf{b}^{\text{dec}} + \sum_{i=1}^{F} f_i(\mathbf{x})\mathbf{W}^{\text{dec}}_{:,i}, \tag{12}$$

where $\mathbf{W}^{\text{dec}} \in \mathbb{R}^{D \times F}$ are the decoder weights.

**Loss Function.**  The SAE is trained by minimizing a loss function $\mathcal{L}$ that typically combines a reconstruction error term and an L1 sparsity penalty on the feature activations:

$$\mathcal{L} = \mathbb{E}_{\mathbf{x}}\left[\|\mathbf{x} - \widehat{\mathbf{x}}\|_2^2 + \lambda \mathcal{L}_1\right], \tag{13}$$

where $\lambda$ is a hyperparameter controlling the strength of the sparsity penalty. The L1 penalty term, $\mathcal{L}_1$, can take different forms depending on constraints applied to the decoder weights:

$$\mathcal{L}_1 = \begin{cases} \sum_{i=1}^{F} |f_i(\mathbf{x})| & \text{(if decoder weights } \mathbf{W}^{\text{dec}}_{:,i} \text{ are unit-normalized)}, \\ \sum_{i=1}^{F} |f_i(\mathbf{x})| \cdot \|\mathbf{W}^{\text{dec}}_{:,i}\|_2 & \text{(if no unit norm constraint on decoder weights)}. \end{cases} \tag{14}$$

In our experiments, we follow the setup where decoder weights are unit-normalized during training, thus using the first form of the L1 penalty.

## C  DETAILED DERIVATION OF THE TRAINING-STEP CONTINUITY THEOREM

Assume the conditions hold. Using a first-order Taylor expansion:

$$\mathbf{x}^{(l,t)} \approx \mathbf{x}^{(l,t-1)} + \left.\frac{\partial F^{(l)}}{\partial \Theta^{(<l)}}\right|_{\Theta^{(<l,t-1)}} \cdot (\Theta^{(<l,t)} - \Theta^{(<l,t-1)}). \tag{15}$$

Substituting the gradient descent update $\Theta^{(<l,t)} - \Theta^{(<l,t-1)} = -\eta \nabla_{\Theta^{(<l)}} \mathcal{L}(\Theta^{(t-1)})$, we have:

$$\mathbf{x}^{(l,t)} \approx \mathbf{x}^{(l,t-1)} - \eta \left.\frac{\partial F^{(l)}}{\partial \Theta^{(<l)}}\right|_{\Theta^{(<l,t-1)}} \cdot \nabla_{\Theta^{(<l)}} \mathcal{L}(\Theta^{(t-1)}). \tag{16}$$

Taking norms and applying the bounds on $\frac{\partial F^{(l)}}{\partial \Theta^{(<l)}}$ and $\nabla_{\Theta^{(<l)}} \mathcal{L}(\Theta)$:

$$\left\|\mathbf{x}^{(l,t)} - \mathbf{x}^{(l,t-1)}\right\| \leq \eta LG. \tag{17}$$

With $\eta < \frac{\epsilon}{LG}$, this ensures:

$$\left\|\mathbf{x}^{(l,t)} - \mathbf{x}^{(l,t-1)}\right\| < \epsilon, \tag{18}$$

proving continuous activation changes over training steps.

This derivation supports the Training-Step Continuity Theorem by bounding activation changes through Lipschitz continuity and gradient norms. The result highlights the incremental and stable evolution of activations during training.

## D  FEATURE SPACE METRIC

**Definition of Feature Space Progress Measure.**  To capture a finer-grained perspective on feature formation, we extend the progress measure to the feature space itself. Given that each SAE-encoded feature captures a specific semantic representation, we define the feature space metric $M_i^{\text{feature}}(t)$ as:

$$M_i^{\text{feature}}(t) = \overline{\text{Sim}}_{\mathcal{F}_i^t} - \overline{\text{Sim}}_{\mathcal{F}_{\text{random}}}, \tag{19}$$

where:

- $\mathcal{F}_i^t$ is the set of feature vectors for the top-$k$ most activating datapoints for feature $i$ at step $t$,
- $\mathcal{F}_{\text{random}}$ is a set of feature vectors corresponding to randomly sampled, semantically unrelated datapoints, and
- $\overline{\text{Sim}}_{\mathcal{F}_i^t}$ and $\overline{\text{Sim}}_{\mathcal{F}_{\text{random}}}$ are the average pairwise similarities within the respective sets, defined as:

$$\overline{\text{Sim}}_{\mathcal{F}_i^t} = \frac{2}{|\mathcal{F}_i^t|(|\mathcal{F}_i^t|-1)} \sum_{\substack{f_k, f_j \in \mathcal{F}_i^t \\ j < k}} \text{Sim}(f_k, f_j), \tag{20}$$

$$\overline{\text{Sim}}_{\mathcal{F}_{\text{random}}} = \frac{2}{|\mathcal{F}_{\text{random}}|(|\mathcal{F}_{\text{random}}|-1)} \sum_{\substack{f_k, f_j \in \mathcal{F}_{\text{random}} \\ j < k}} \text{Sim}(f_k, f_j). \tag{21}$$

This formulation extends the datapoint-level analysis to the underlying feature vectors themselves, capturing the degree to which a feature stabilizes within its high-dimensional embedding space as training progresses.

## E  DEAD FEATURES AND ULTRA-LOW ACTIVATION FEATURES

In our analysis, we exclude two categories of features that fail to contribute meaningful information during training:

- **Dead Features**: These are features that do not activate on any datapoint. Such features are entirely uninformative and irrelevant to the our study.
- **Ultra-Low Activation Features**: Features with extremely low activation densities or values are also excluded. While not strictly inactive, these features exhibit negligible activations that render them semantically meaningless. This filtering is consistent with prior observations in (Bricken et al., 2023), which identify such low-activation features as non-contributive.

By filtering these two types of features, we focus on those that exhibit meaningful activations and contribute to the evolving structure of the activation space, enabling a clearer study of feature dynamics.

## F  WEAK CONCEPT-LEVEL FEATURES

Concept-level features with limited variants, such as morphological features corresponding to suffixes (e.g., *-ed*, *-ing*), can be considered weak concept-level features. For instance, a feature might primarily activate for 10 occurrences of *-ing* and 15 of *-ed*, leading to repeated pairings during similarity calculations. This repetition often inflates similarity scores in similarity-based metrics, despite these features being fundamentally identical in nature to typical concept-level features.

Table 2: **Weak Concept-Level Features.**

| | Feature Activation |
|---|---|
| **Weak Concept-Level Features** 4/5464 | 1. being utilized in mass drug administration campaigns
2. by utilizing SEO tips for beginners
3. utilizing toys around the house like ...
4. may be selected and utilized to record other bodily ionic ...
5. Shelton then utilized a technique whereby ...
6. so that it may be utilized by the specific print engine
7. develop and utilize the seven ESI skills
8. jargon will be utilized which is identified ...
9. to the procedure to be utilized in determining ...
10. most processes have utilized the blade outer ...
11. it is utilized by the over 1, ...
12. the system can be utilized to significantly ...
13. likely to be utilized in the arithmetic unit
14. if you aren't utilizing social networking ...
15. are utilized by all levels of fly ...
16. to make their websites utilized more effectively
17. help your employees choose and utilize their benefits
18. fans on all headers utilize high-amperage
19. none of it utilized multiple threads ...
... |

At the start of training, datapoints corresponding to weak features often form multiple separate clusters in the activation space. However, this clustering is a superficial phenomenon that reflects redundancy rather than meaningful semantic coherence. Only via training does the model gradually learn to organize these datapoints into a single cohesive feature.

## G  POLYSEMOUS TOKEN-LEVEL FEATURES.

For polysemous tokens – tokens with multiple meanings – the corresponding token-level features may initially activate without capturing any semantic distinctions. During the early training phase, these features are primarily activated based on token identity alone. However, as training progresses and the model learns to incorporate semantics, these features sometimes degrade to represent only the most prominent meaning of the token. This degradation reflects the model's learning process, where it begins to understand and refine what a token-level feature truly represents, prioritizing the most frequent or contextually significant meaning.

Table 3: **Polysemous Token-Level Features.** Words in blue denote "resolute" or "solid", while green indicates "company" or "organization", highlighting the model's refinement of polysemous meanings during training.

| | checkpoint 0 | checkpoint 15–21 | checkpoint 153 (final) |
|---|---|---|---|
| **Degradation of Polysemy** 4/20242 | 1. hold firm and cherish
2. issued a firm threat
3. oil firm for decades
4. absolutely firm and ...
5. landscape design firm
6. the US firm is not... | 1. the US firm is not...
2. brokerage firm CPNA
3. landscape design firm
4. no organization or firm ...
5. as a firm, purple-blue...
6. have a firm mattress | 1. North Carolina-based firm
2. prestigious law firm
3. the firm offers probate
4. from the law firm
5. policy of the firm
6. by leading US law firm, |

# H    THE CHALLENGE OF TRACKING THE SEMANTICS OF INITIAL FEATURES

One might expect that all (token-level) features observed during the initialization stage can be consistently tracked throughout training. However, this is not feasible due to several key reasons:

- **Emergent Phase Dynamics**: During the emergent phase, activations corresponding to initially distinct datapoints may overlap or merge, resulting in features that no longer align with their initial definitions.

- **Lack of Ideal Continual Steps**: There are no ideal continual training steps, as discussed in Theorem 2.1, which complicates the tracking of features across training, especially in stages where the checkpoints are sparsely distributed.

- **SAE Training Property**: SAE training can be viewed as selecting features from a large pool of possible features to explain the model activations (Templeton et al., 2024). Even when training SAEs twice on the same model activations and data, divergence in learned features can occur (Bricken et al., 2023). This selection process inherently introduces inconsistencies between initial and final features.

- **Shifting Phenomenon**: Unlike the initial checkpoint, where SAEs mainly produce token-level features, the final checkpoint SAEs are not constrained to token-level representations. As training progresses, features initially aligned to specific tokens may shift and evolve into other features. This transformation makes strict feature tracking across checkpoints impractical.

- **The Impact of Possible Feature Collapse:** See Discussion Appendix L.2 at Appendix L.

It is important to emphasize that SAE-Track is designed as a study tool rather than an engineering evaluation framework. The goal is to provide insights into feature dynamics, not to enforce strict feature-tracking consistency.

# I    IMPLEMENTATION DETAILS

Most experiments were conducted on a single NVIDIA A100 GPU. The implementation is built primarily upon open-source codebases (McDougall, 2024; Joseph Bloom & Chanin, 2024; Lin, 2024; Nanda & Bloom, 2022).

**Models and Datasets:** We use the Pythia-deduped models (Biderman et al., 2023) and Stanford CRFM Mistral (Karamcheti et al., 2021) for our experiments.

Pythia provides 154 checkpoints across training, with the checkpoints recorded at the following training steps:

$$[0, 1, 2, 4, 8, 16, 32, 64, 128, 256, 512] + \text{list}(\text{range}(1000, 143000 + 1, 1000)).$$

We conduct experiments on three Pythia scales: 160M, 410M, and 1.4B parameters, ensuring consistency across model sizes.

Stanford CRFM Mistral (Karamcheti et al., 2021) provides an open-source replication of the GPT-2 model (Radford et al., 2019), including five GPT-2 Small and five GPT-2 Medium models trained on the OpenWebText corpus (Gokaslan et al., 2019). Each model produces 609 checkpoints, recorded at the following training steps:

$$\text{list}(\text{range}(0, 100, 10)) + \text{list}(\text{range}(100, 2000, 50))$$
$$+ \text{list}(\text{range}(2000, 20000, 100)) + \text{list}(\text{range}(20000, 400000 + 1, 1000)).$$

Datasets used in training SAE-Track and conducting mechanistic experiments correspond to the datasets used during model training.

Stanford CRFM GPT-2 models use the OpenWebText corpus (Gokaslan et al., 2019), while Pythia models use the deduplicated version of the Pile dataset (Gao et al., 2020). The deduplicated Pile dataset ensures minimal repetition in the training data, aligning with the Pythia-deduped models.

**SAEs Training:** To efficiently train SAEs across multiple checkpoints, we employ a recurrent initialization scheme, which reuses the weights from the previous checkpoint to initialize the current SAE. The checkpoints for SAE training are selected based on an adaptive schedule:

$$S = \bigcup_{i=1}^{M} \bigcup_{j=0}^{n_i-1} (a_i + d_i \cdot j), \quad \text{where} \begin{aligned} &M \text{ is the total number of segments,} \\ &a_i \text{ is the starting value of the } i\text{-th segment,} \\ &d_i \text{ is the step size of the } i\text{-th segment,} \\ &n_i \text{ is the number of elements in the } i\text{-th segment,} \\ &a_i = a_{i-1} + d_{i-1} \cdot n_{i-1} \text{ ensures continuity.} \end{aligned} \tag{22}$$

This piecewise linear schedule adapts the checkpoint density across training phases. In later training checkpoints, fewer checkpoints suffice as feature evolution slows, reducing computational costs while preserving representation quality. However, using all checkpoints or a denser selection can enhance tracking precision when needed.

Overtraining is applied to enhance feature representations, as recommended in (Bricken et al., 2023). By leveraging the recurrent initialization scheme, which reuses pretrained weights, convergence is significantly accelerated. Specifically, only $\leq \frac{1}{20}$ of the initial training tokens are required for subsequent SAEs, resulting in substantial computational savings.

## I.1 COMPARATIVE STUDY: SAE-TRACK VS. CONVENTIONAL SAE TRAINING

**A Training Example:** Here we present an example trained on **Pythia-410M-Deduped**.

We follow the training schedule:

$$\text{list(range(33))} + \text{list(range(33, 153, 5))} \tag{23}$$

SAE[0] is trained using 300M tokens, while checkpoints 1–4 each use 5M tokens, and all remaining checkpoints use 15M tokens. The training details are illustrated in terms of overall loss, MSE loss, explained variance, and $L_0$ metric.

For comparison, we also train an SAE on the final LLM checkpoint, following the commonly used approach of training SAEs on a fixed checkpoint. The results show that SAE-Track-generated SAEs exhibit similar behavior to normally trained SAEs, but converge significantly faster.

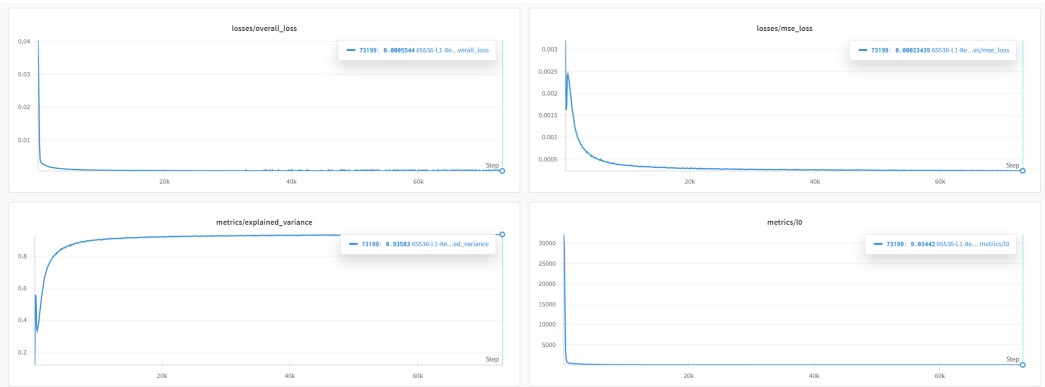

Figure 7: SAE[0] training, SAE-Track

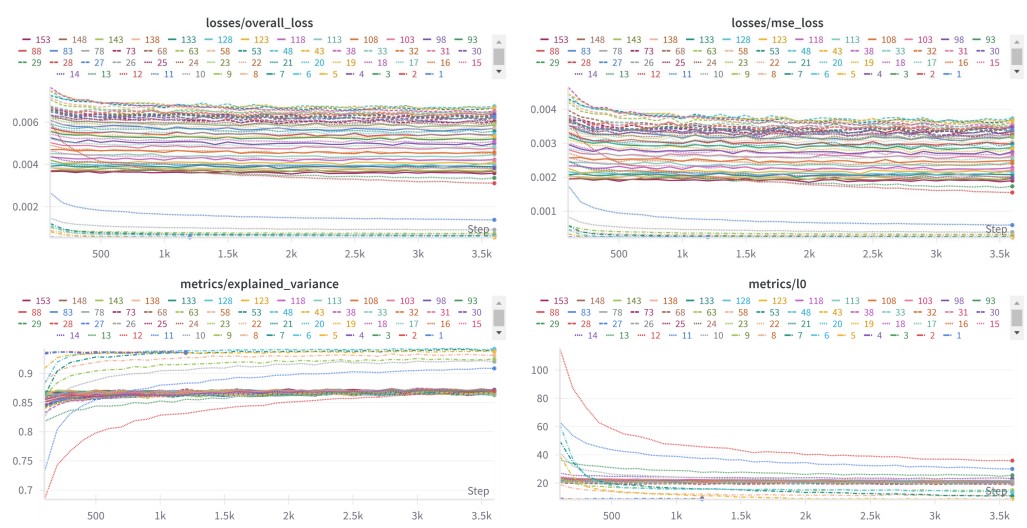

Figure 8: SAE[1]-SAE[153] training, SAE-Track

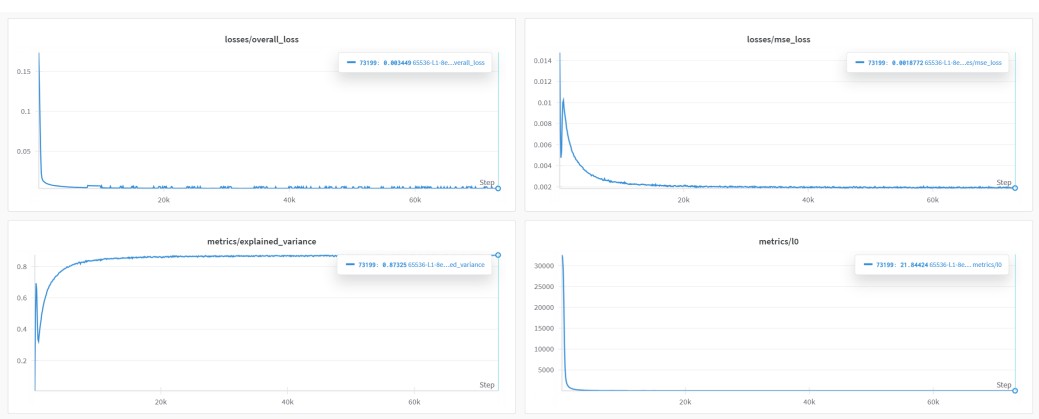

Figure 9: SAE trained (normally) on 153

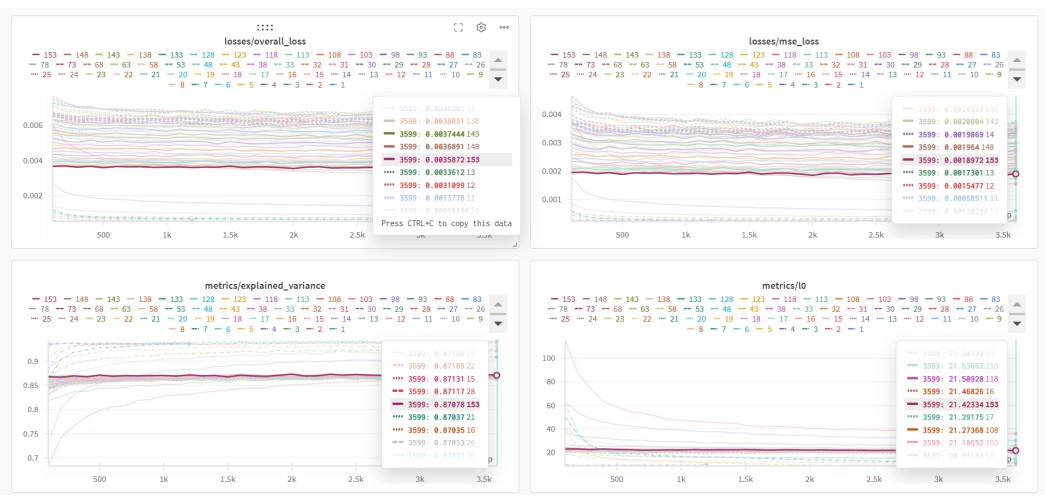

Figure 10: SAE[153](SAE-Track) share similar behavior of normally trained SAE on checkpoint 153

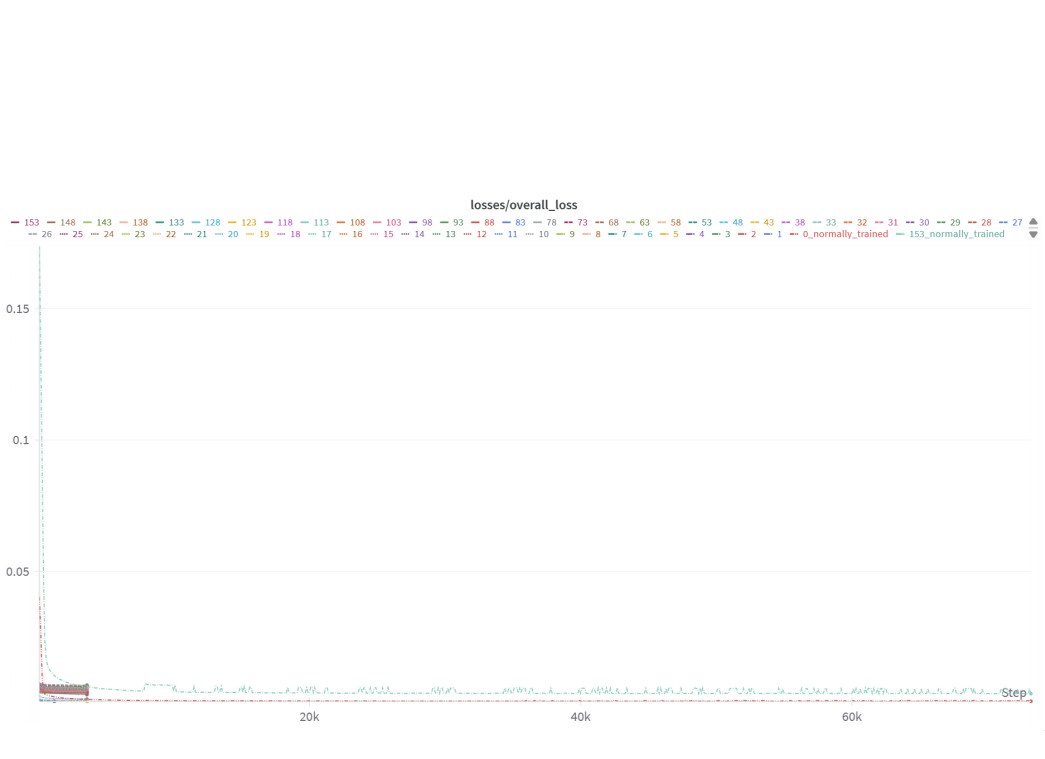

Figure 11: Comparing converging speed between SAE-Track and normal training

## I.2 COMPARATIVE STUDY: REVERSE TRACKING VS. FORWARD TRACKING

To further validate the robustness and consistency of SAE-Track, we conducted a reverse tracking experiment. Unlike the standard forward training, where each SAE is initialized from the previous checkpoint, this approach starts from the final SAE and progressively finetunes backward through earlier checkpoints. This design aims to evaluate whether the observed feature convergence in SAE-Track is a genuine phenomenon or merely an artifact of its forward-only training strategy.

Specifically, the reverse tracking experiment addresses the concern that early-stage SAEs might prematurely consume the available capacity, limiting the ability of later stages to incorporate newly emerging, complex features. By reversing the training direction, we can test whether the observed convergence is genuinely a feature of the underlying data distribution and model architecture, rather than an unintended consequence of the forward training process.

Figures 12 and 13 present the results of this reverse tracking analysis. Despite reversing the training direction, we observe that the overall feature formation and alignment remain consistent, suggesting that the convergence observed in forward training is not solely a byproduct of incremental parameter freezing, but a more fundamental property of the model's feature dynamics.

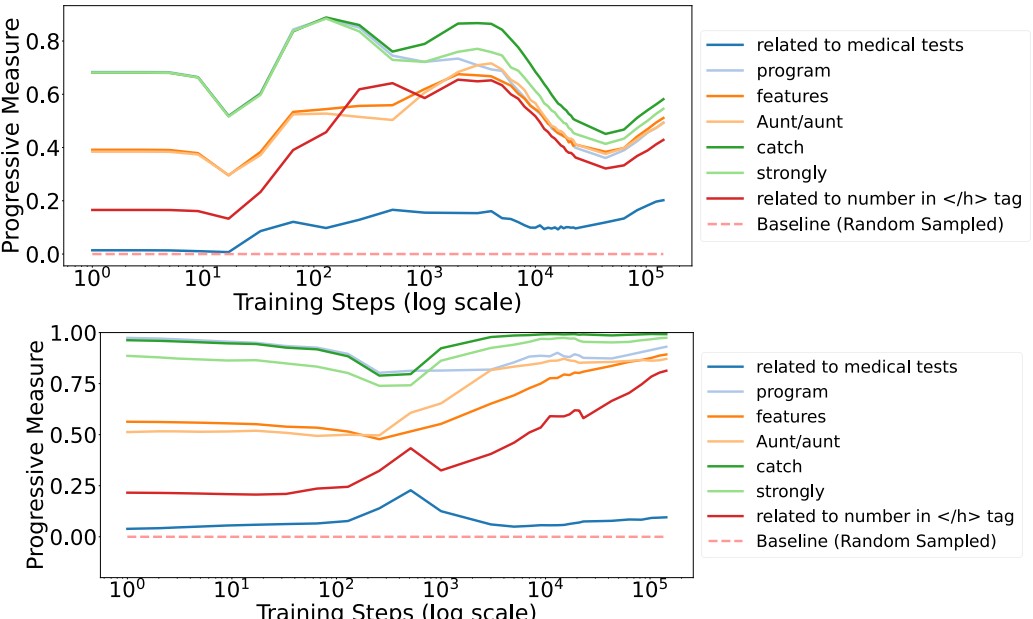

Figure 12: Progress Measure for reverse tracking. The top panel shows the feature space, while the bottom panel represents the activation space. Despite reversing the training sequence, the overall progression of feature formation remains consistent, indicating the stability and robustness of SAE-Track across training directions.

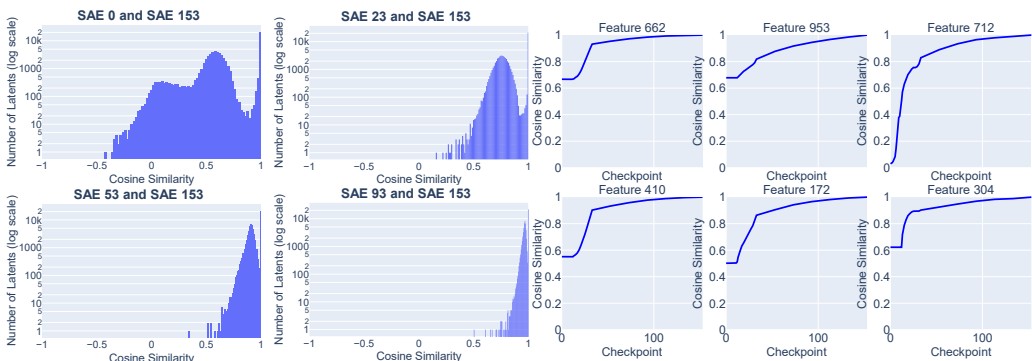

Figure 13: Decoder Cosine Similarity for reverse tracking. Each panel shows the alignment of decoder vectors across training checkpoints in reverse order, confirming that feature direction stability is preserved even when the training order is reversed.

## J   DIFFERENT SIMILARITY METRICS

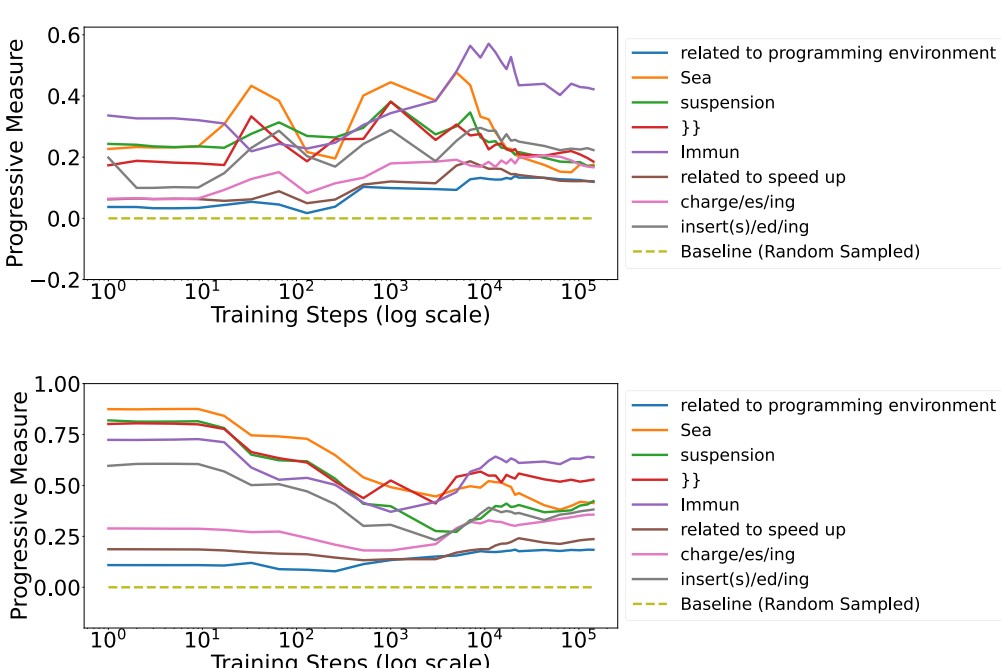

Figure 14: **Progress Measure using different similarity metrics.** Top: jaccard similarity for feature space, Bottom: weighted jaccard similarity for feature space. Conducted on **Pythia-410-Deduped**.

Our progress measure relies on the choice of similarity metrics. In the main text, we use cosine similarity; here, we extend the analysis by exploring additional metrics, as shown in figure 14. The results demonstrate that the overall trend remains consistent across different metrics. Specifically, token-level features exhibit relatively stable high values, while concept-level features gradually increase in similarity metric values as training progresses. Importantly, the choice of similarity metric does not significantly affect the overall analysis or conclusions.

**Definitions of Similarity Metrics:**

- **Cosine Similarity:** Cosine similarity, applied to the activation space with new datapoints, measures the angular similarity between two vectors $\mathbf{u}$ and $\mathbf{v}$. It is defined as:

$$\text{CosSim}(\mathbf{u}, \mathbf{v}) = \frac{\mathbf{u} \cdot \mathbf{v}}{\|\mathbf{u}\| \|\mathbf{v}\|}, \tag{24}$$

where $\mathbf{u} \cdot \mathbf{v}$ denotes the dot product, and $\|\mathbf{u}\|$, $\|\mathbf{v}\|$ are the norms of the respective vectors.

- **Jaccard Similarity:** Jaccard similarity is applied to the sparse feature space. It converts each feature vector into a binary representation, indicating whether a feature is activated (1) or not (0), and calculates similarity as:

$$\text{Jaccard}(\mathbf{u}, \mathbf{v}) = \frac{|\mathbf{u}_{\text{binary}} \cap \mathbf{v}_{\text{binary}}|}{|\mathbf{u}_{\text{binary}} \cup \mathbf{v}_{\text{binary}}|}, \tag{25}$$

where $\mathbf{u}_{\text{binary}}$ and $\mathbf{v}_{\text{binary}}$ are the binary representations of $\mathbf{u}$ and $\mathbf{v}$, respectively.

- **Weighted Jaccard Similarity:** Weighted Jaccard similarity extends Jaccard similarity by considering the magnitude of activations in the feature space. For two activation vectors $\mathbf{u}$ and $\mathbf{v}$, it is defined as:

$$\text{WeightedJaccard}(\mathbf{u}, \mathbf{v}) = \frac{\sum_i \min(u_i, v_i)}{\sum_i \max(u_i, v_i)}, \tag{26}$$

where $u_i$ and $v_i$ are the activation values for feature $i$ in $\mathbf{u}$ and $\mathbf{v}$, respectively.

Since Jaccard and Weighted Jaccard are more suitable for sparse vectors, and their meaning becomes less significant for non-sparse vectors, we restrict their use to the feature space. The overall trends presented in figure 14 demonstrate that the choice of metric does not substantially affect the study's conclusions.

## K    EXPERIMENTS ON DIFFERENT MODELS AND LAYERS

### K.1    PYTHIA OF OTHER SCALES

Below, we present results for **Pythia-160m-deduped, layer=4** and **Pythia-1.4b-deduped, layer=3**, trained on the residual stream before the specified layers. The figures include UMAP, progress measures, decoder cosine similarity, and trajectory analysis. These results align closely with those observed for **Pythia-410m-deduped, layer=4** in the main paper, highlighting the consistency of our results.

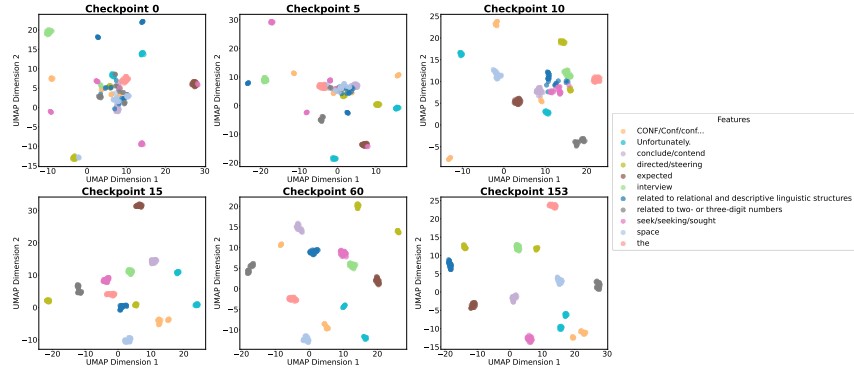

Figure 15: **UMAP for Pythia-160m-deduped.**

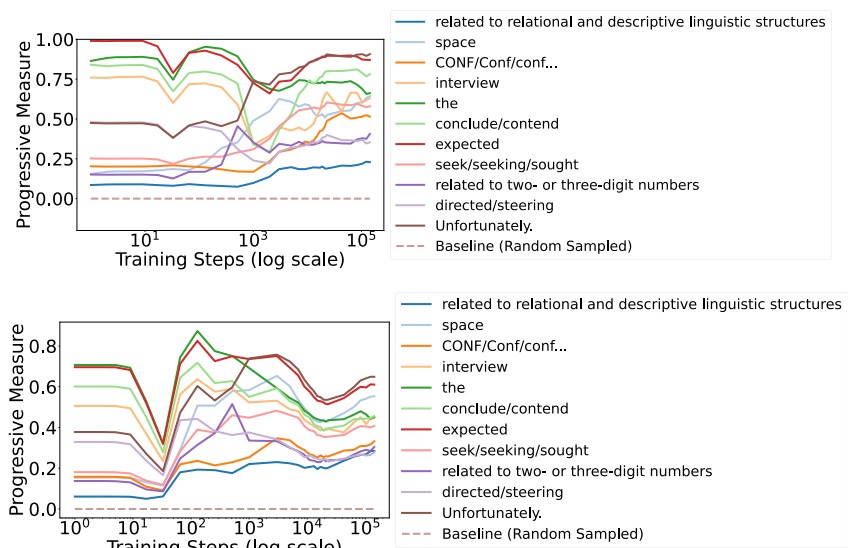

Figure 16: **Progress Measure for Pythia-160m-deduped.** The top represents the feature space, while the bottom represents the activation space.

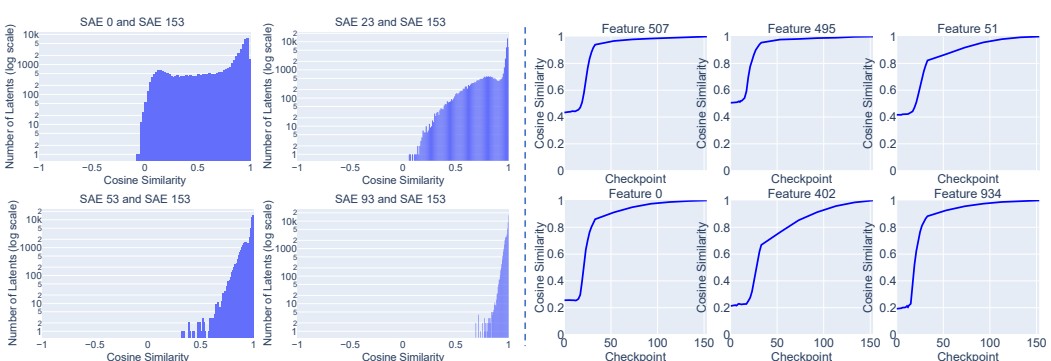

Figure 17: **Cosine Similarity for Pythia-160m-deduped.**

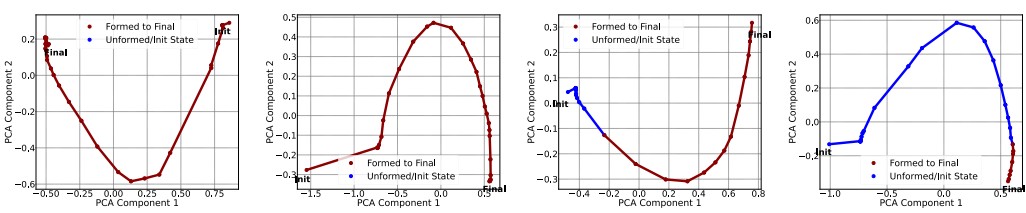

Figure 18: **Feature Trajectories for Pythia-160m-deduped.**

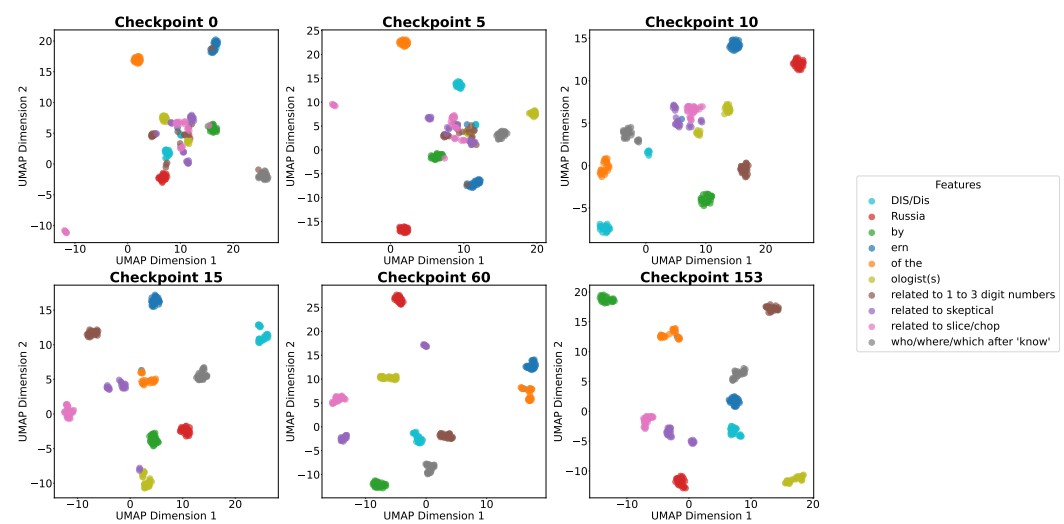

Figure 19: **UMAP for Pythia-1.4b-deduped.**

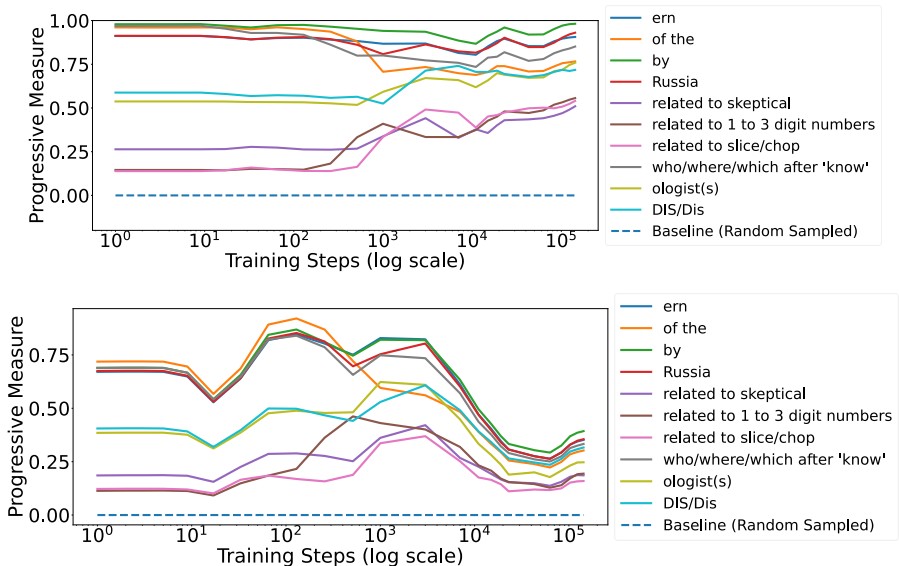

Figure 20: **Progress Measure for Pythia-1.4b-deduped.** The top represents the feature space, while the bottom represents the activation space.

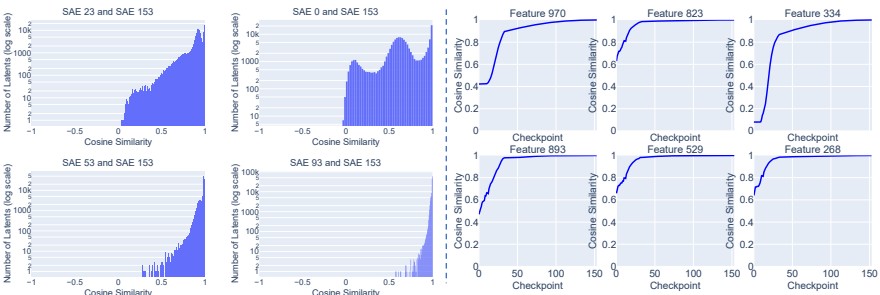

Figure 21: **Cosine Similarity for Pythia-1.4b-deduped.**

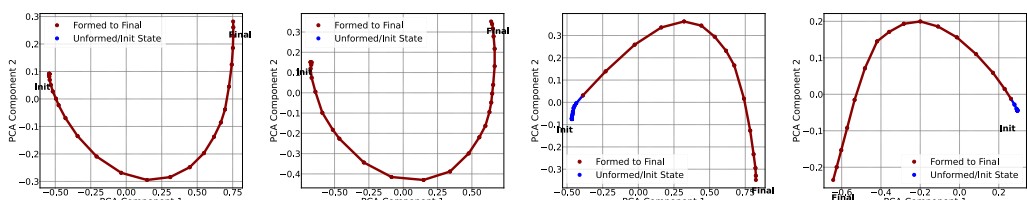

Figure 22: **Feature Trajectories for Pythia-1.4b-deduped.**

## K.2 STANFORD GPT2 OF DIFFERENT SCALES

Below, we present results for **stanford-gpt2-small-a, layer=5** and **stanford-gpt2-medium-a, layer=6**, trained on the residual stream before the specified layers. The figures include UMAP, progress measures, decoder cosine similarity, and trajectory analysis.

The results are mainly consistent, except for the glitch observed in Stanford-GPT2-Small-A and the UMAP of initialization. The UMAP at initialization appears more diverged, which is related to both the initialization scheme and the model architecture. However, token-level features still exist at this stage. The glitch is further explained in Appendix L.

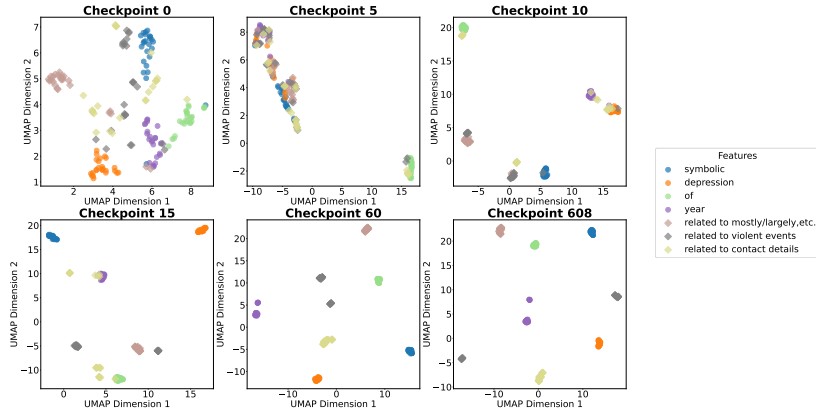

Figure 23: **UMAP for stanford-gpt2-small-a.**

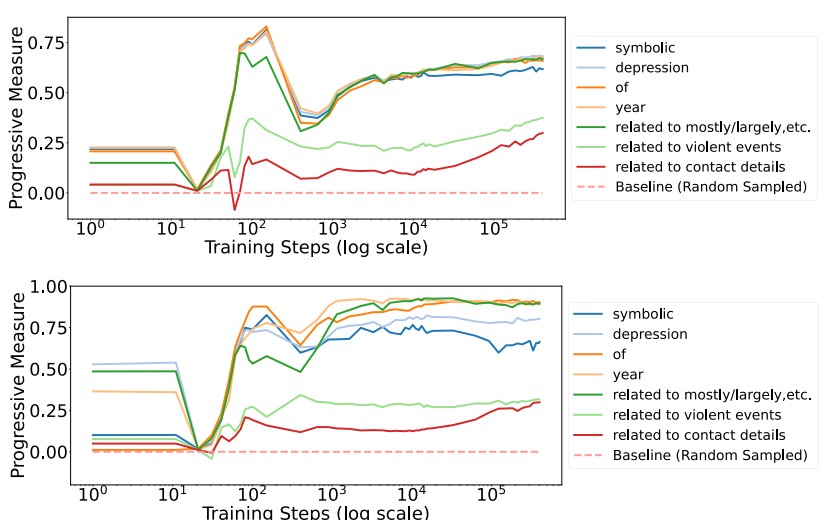

Figure 24: **Progress Measure for stanford-gpt2-small-a.** The top represents the activation space, while the bottom represents the feature space.

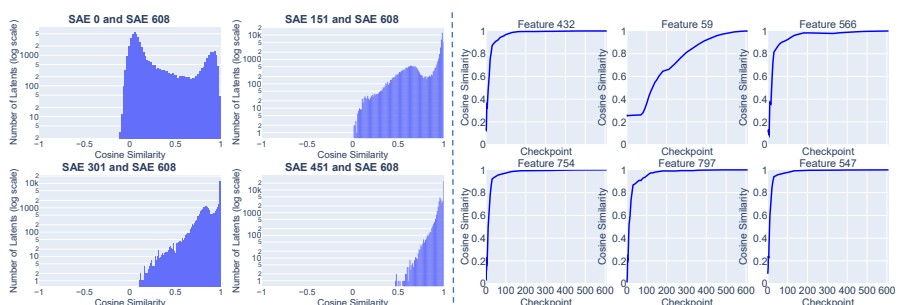

Figure 25: **Cosine Similarity for stanford-gpt2-small-a.**

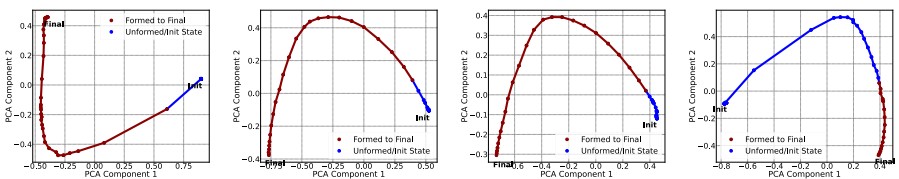

Figure 26: **Feature Trajectories for stanford-gpt2-small-a.**

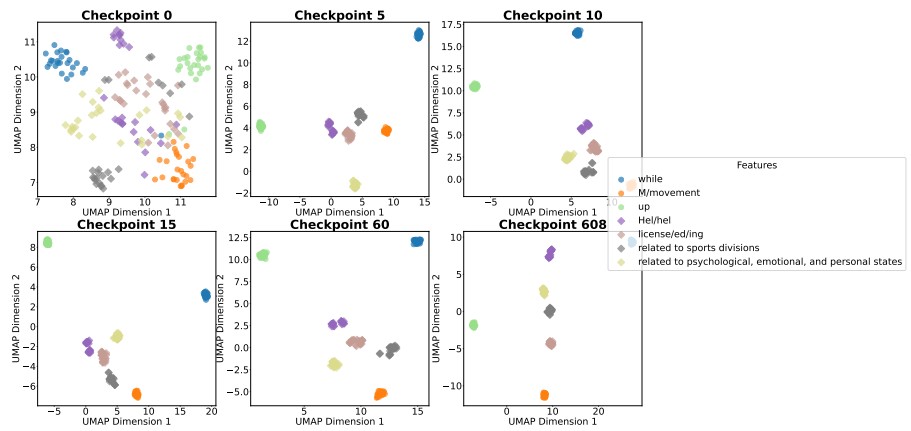

Figure 27: **UMAP for stanford-gpt2-small-a.**

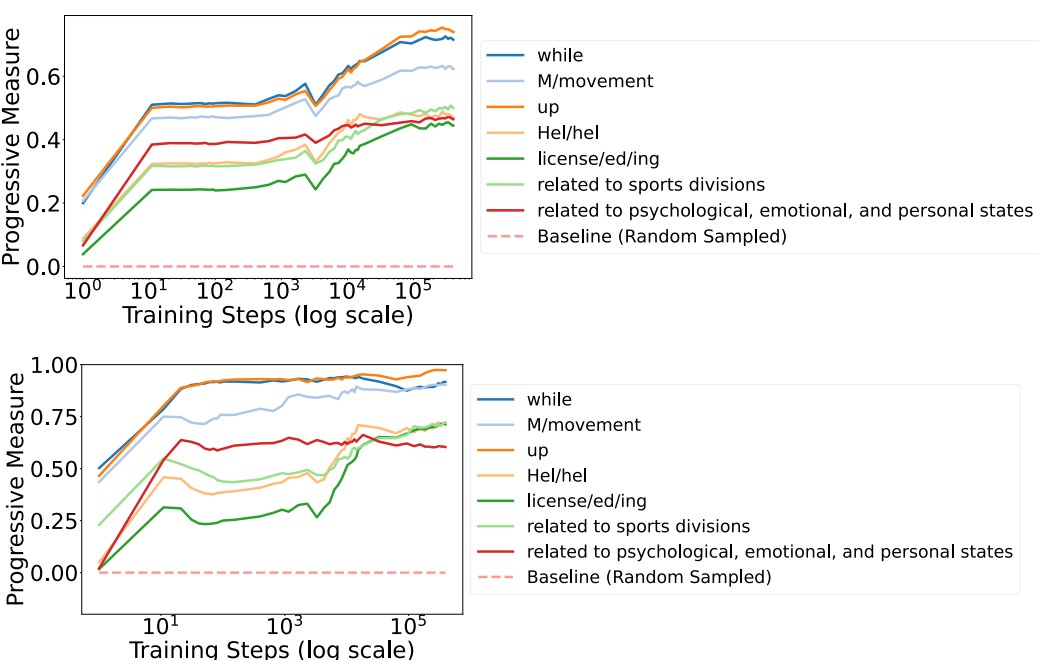

Figure 28: **Progress Measure for stanford-gpt2-medium-a.** The top represents the activation space, while the bottom represents the feature space.

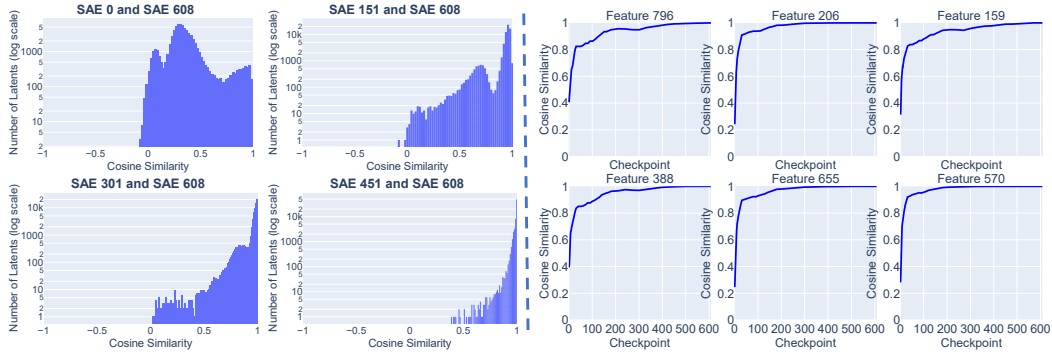

Figure 29: **Cosine Similarity for stanford-gpt2-medium-a.**

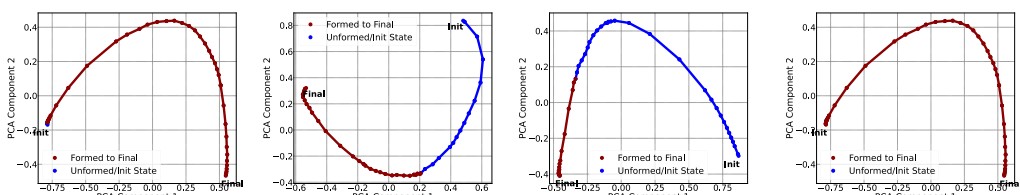

Figure 30: **Feature Trajectories for stanford-gpt2-medium-a.**

### K.3 Experiments on Different Layers, Pythia-410m-deduped

In this section, we present experiments on different layers of the Pythia-410m-deduped model. These experiments aim to capture the feature formation and alignment dynamics across various layers, providing insights into how layer depth influences feature specialization and semantic coherence. Specifically, we include both the Progress Measure and Cosine Similarity analyses for layers 8, 12, 16, and 20. The Progress Measure captures the gradual semantic formation of features in both the activation and feature spaces, while the Cosine Similarity plots reveal the directional alignment of decoder vectors across training steps.

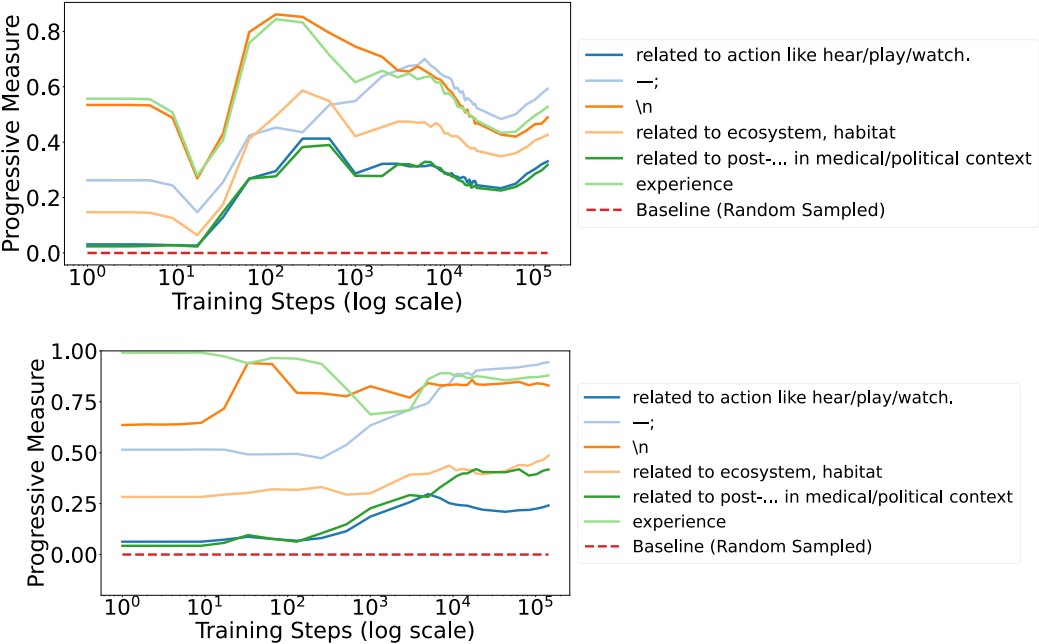

Figure 31: Progress Measure for Pythia-410m-deduped, layer 8. The top panel represents the activation space, while the bottom panel represents the feature space.

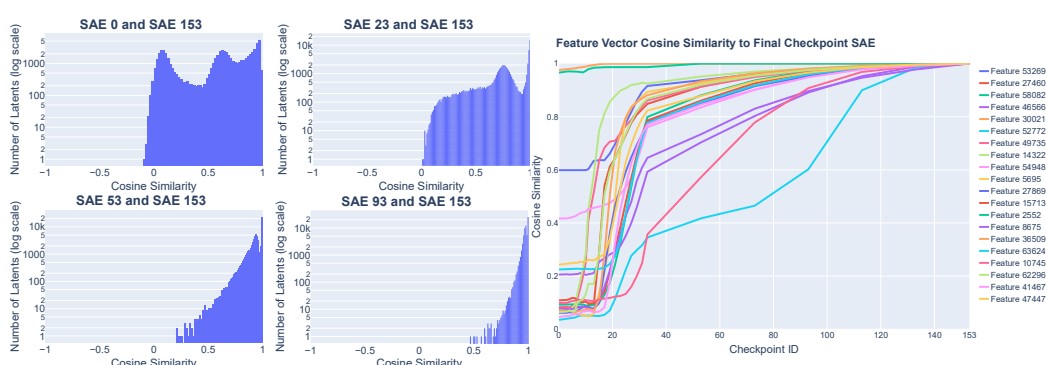

Figure 32: Cosine Similarity for Pythia-410m-deduped, layer 8. Each panel shows the cosine similarity between decoder vectors at different training checkpoints.

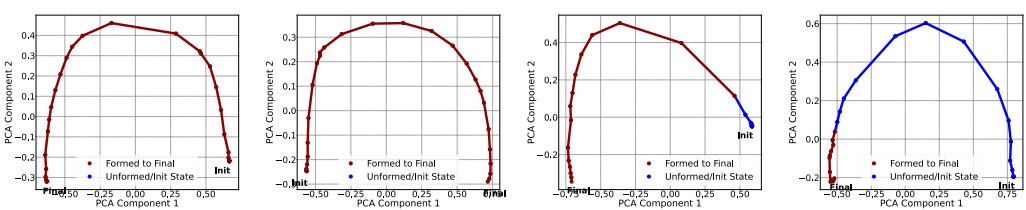

Figure 33: Feature Trajectories for Pythia-410m-deduped, layer 8.

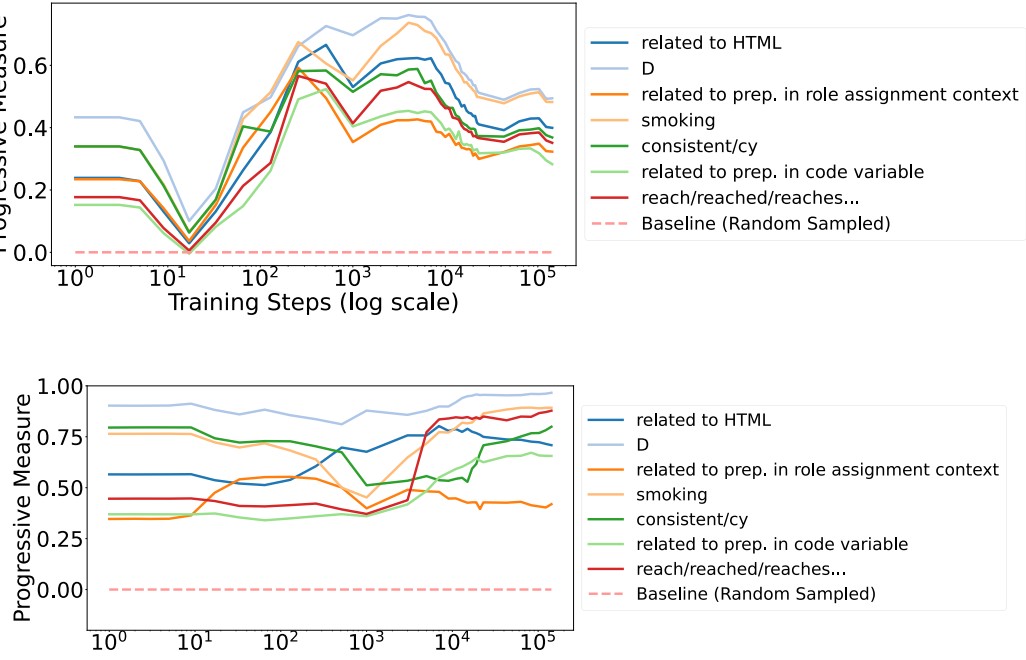

Figure 34: Progress Measure for Pythia-410m-deduped, layer 12. The top panel represents the activation space, while the bottom panel represents the feature space.

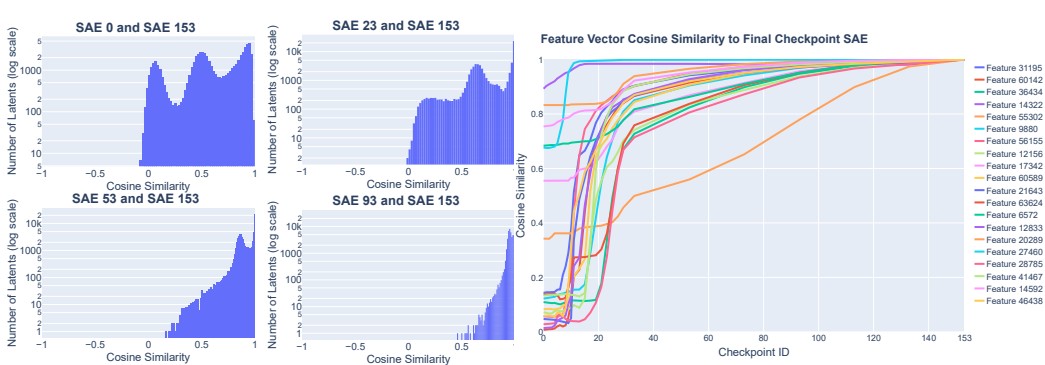

Figure 35: Cosine Similarity for Pythia-410m-deduped, layer 12. Each panel shows the cosine similarity between decoder vectors at different training checkpoints.

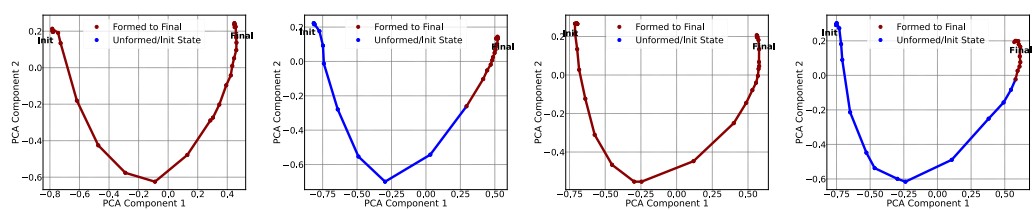

Figure 36: Feature Trajectories for Pythia-410m-deduped, layer 12.

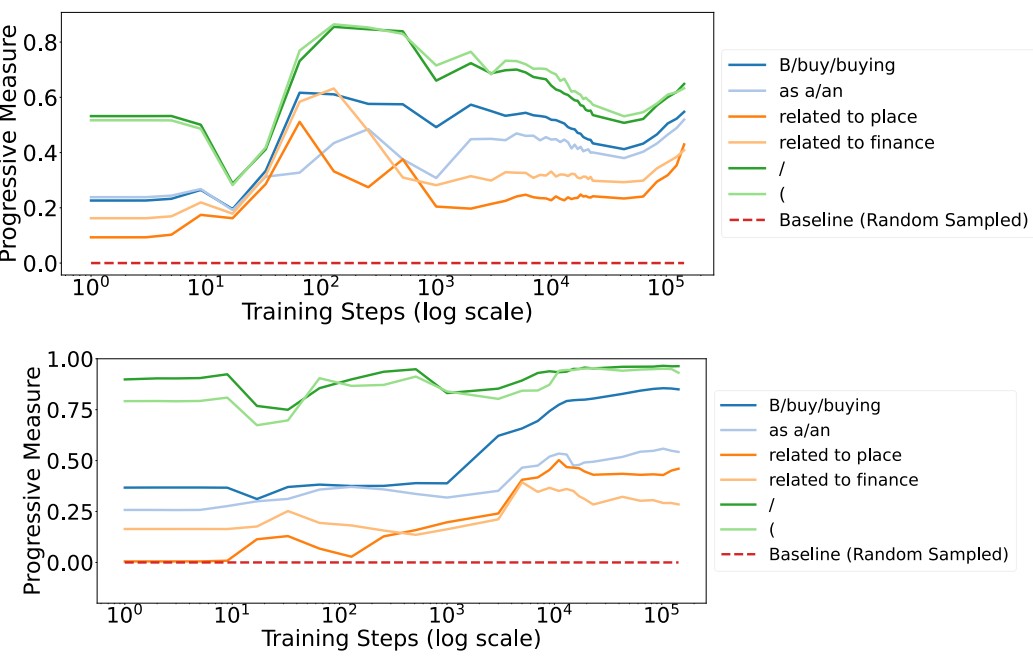

Figure 37: Progress Measure for Pythia-410m-deduped, layer 16. The top panel represents the activation space, while the bottom panel represents the feature space.

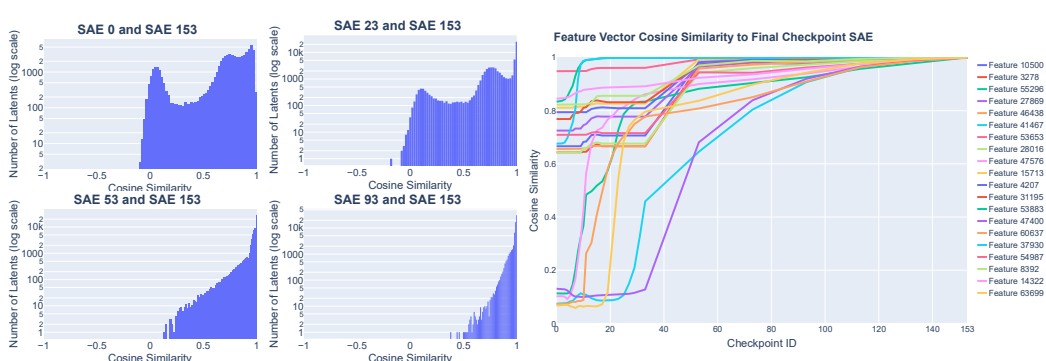

Figure 38: Cosine Similarity for Pythia-410m-deduped, layer 16. Each panel shows the cosine similarity between decoder vectors at different training checkpoints.

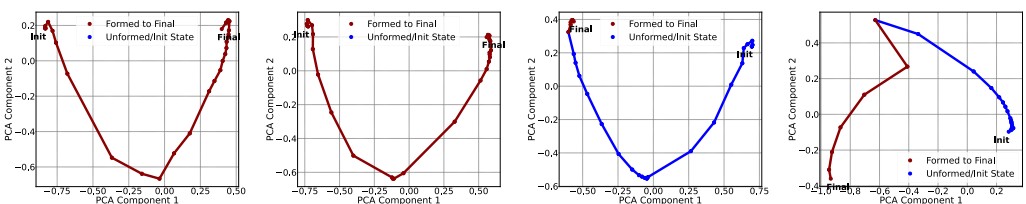

Figure 39: Feature Trajectories for Pythia-410m-deduped, layer 16.

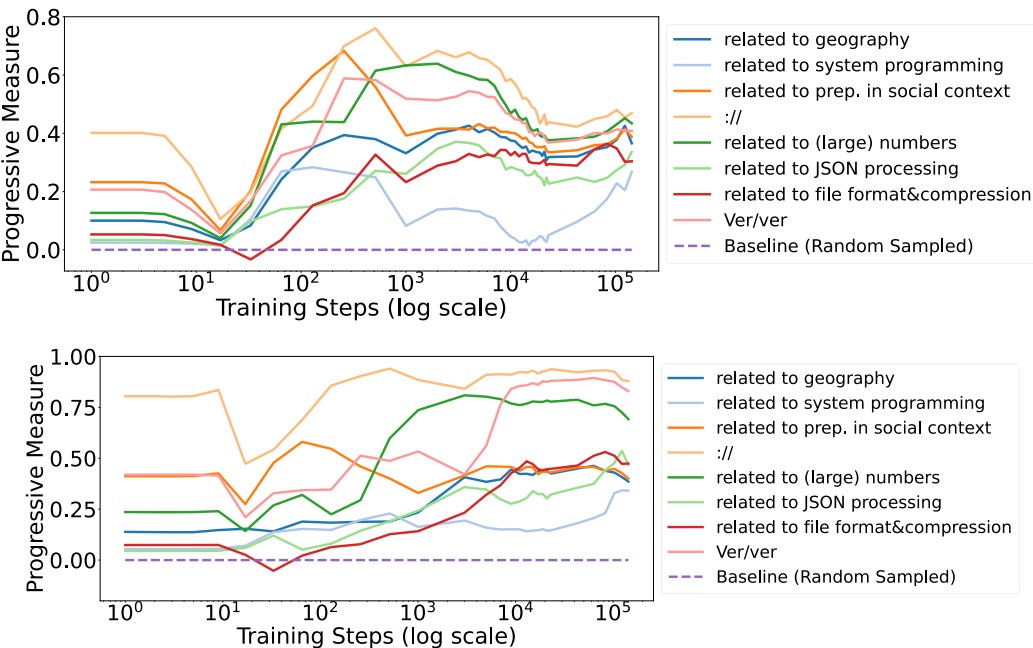

Figure 40: Progress Measure for Pythia-410m-deduped, layer 20. The top panel represents the activation space, while the bottom panel represents the feature space.

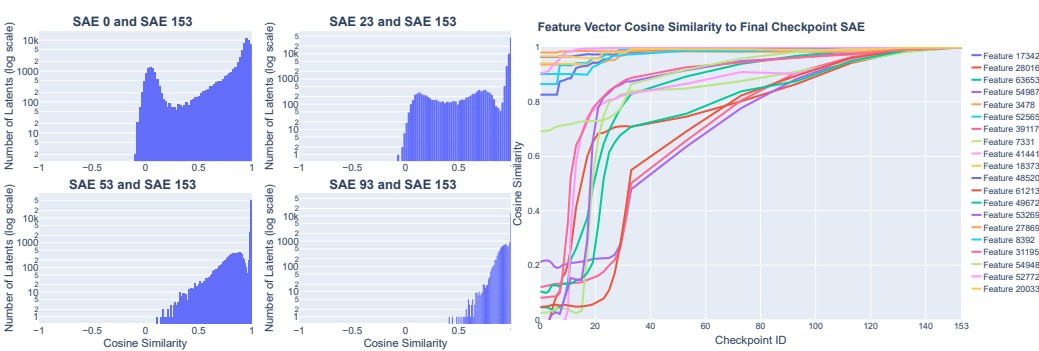

Figure 41: Cosine Similarity for Pythia-410m-deduped, layer 20. Each panel shows the cosine similarity between decoder vectors at different training checkpoints.

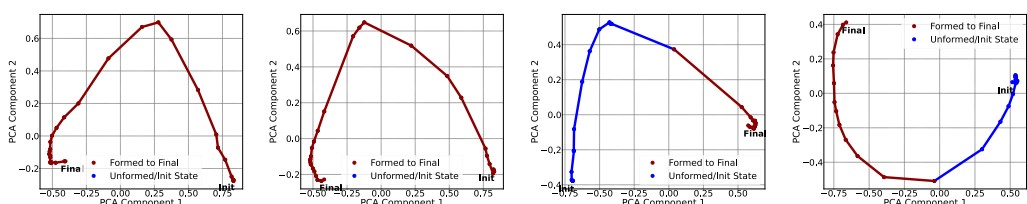

Figure 42: Feature Trajectories for Pythia-410m-deduped, layer 20.

## L  POSSIBLE FEATURE COLLAPSE

We observe a phenomenon we refer to as **feature collapse** in the early checkpoints of certain models (e.g., GPT-2-small-a, where it is most pronounced).

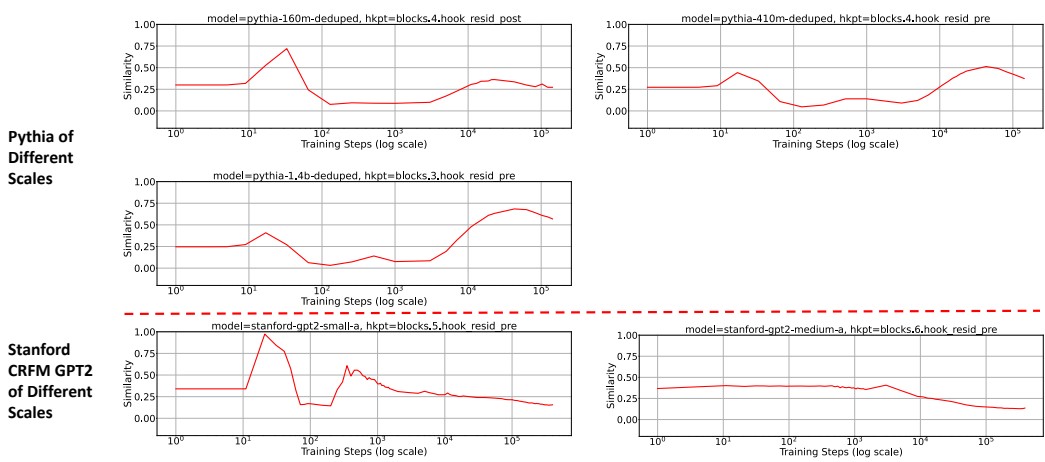

Figure 43: **Feature Collapse**

**Feature collapse**, in this context, refers to a state where all activations or features converge to near-1 cosine similarity. This phenomenon manifests as a glitch in the our measure (figure 4), which we attribute to a sudden burst in $\overline{\text{Sim}}_{\mathcal{A}_{\text{random}}}$ (Figure 43). Specifically, when $\overline{\text{Sim}}_{\mathcal{A}_{\text{random}}}$ approaches 1, such that $1 - \overline{\text{Sim}}_{\mathcal{A}_{\text{random}}} < \epsilon$, and given that $\overline{\text{Sim}}_{\mathcal{A}_i^t} \leq 1$, the measure $M_i(t) = \overline{\text{Sim}}_{\mathcal{A}_i^t} - \overline{\text{Sim}}_{\mathcal{A}_{\text{random}}}$ becomes suppressed to near-zero values ($< \epsilon$), leading to the glitch.

### L.1 MODEL-SPECIFIC PHENOMENON

This collapse is model-specific and may be linked to optimization settings, particularly during early training. Notably, we find that this behavior parallels observations in (Liu et al., 2022), where the cosine similarity of activations (referred to as "features" in their work but analogous to "activations" in our context) increases significantly in early training before decreasing. The observed increase in cosine similarity in our context induces the collapse, as it artificially reduces the ability of any metric to distinguish features.

### L.2 IMPACT ON SAE-TRACK

When feature collapse is severe(where $\overline{\text{Sim}}_{\mathcal{A}_{\text{random}}}$ rapidly approaches 1), it causes disruptions in the tracking process. Specifically, SAE-Track may exhibit phase shifts near the collapse point, reflecting distinct deviations. This is because feature collapse compromises SAE's ability to preserve feature properties, making tracking behavior more unstable and inconsistent.

However, severe feature collapse does not always occur in LLMs (only one of the LLMs we tested exhibited severe feature collapse). When it does occur, it typically happens at a very early point in training, where we still retain a long and compact tracking range for the remaining training process. So SAE-Track remains capable of preserving the majority of feature information and provides complete and accurate tracking beyond this point. By focusing on checkpoints after the collapse, we ensure that the feature trajectories remain stable and interpretable in the subsequent analysis.

