# OpenReview forum: "Tracking the Feature Dynamics in LLM Training: A Mechanistic Study"
_ICLR.cc/2026/Conference — Submitted to ICLR 2026_

### Official Review · Reviewer_d3PW · 2025-10-17

**Soundness:** 3
**Presentation:** 1
**Contribution:** 3
**Rating:** 4
**Confidence:** 3

**Summary:**

The paper investigates the dynamics of SAE features across training by leveraging the transfer learning of SAEs.
They first classify features as concept/token level, and introduce a taxonomy of feature meaning changes.
They then introduce theoretical tooling to investigate shifts across training, show this with UMAP, and introduce a progress measure.
Their work indicates that features evolve across 3 phases, and that they can continue to shift geometrically even after their semantic meaning is established.

**Strengths:**

Showing that token-level features appear from the start is very interesting, because it means SAEs can learn seemingly meaningful features out of activations of random networks.

Introducing a taxonomy of feature transitions is helpful, and can be built upon by further work.
Definition 4.1 is very useful for future work, and so if the UMAP visualisation (as an illustration).

The 3 phases of feature evolution are an interesting finding, and supported with quantitative experiments and insightful new theoretical devices (ex. def 4.2)

The fact that interpretable features can continue to geometrically shift is an interesting (counterintuitive) empirical insight.

**Weaknesses:**

SAE transfer learning (called recurrent initialisation in the paper and presented as a core contribution), was introduced last year in this paper: https://aclanthology.org/2024.blackboxnlp-1.32/, which was not cited.
The work should be cited, and ideally the contributions rescoped (for example from "we introduce recursive initialisation" to "we apply recursive initialisation to model checkpoints").

Other problems:
- feature case studies are mostly used instead of samples of (concept/token level) features.
- no study across model scale (ex. on larger models)

If all (or at least the first 2) weaknesses were address, I could further increase my score to 6 or 8.

**Questions:**

Is your initial checkpoint (0) completely untrained? If so state this, as it strengthens your work by lending support to the claim I outlined in the beginning of the strengths paragraph.

Have you considered studies on larger models?

Have you considered working with samples of features (with quantitative results) as opposed to select features for case studies?

---

### Official Review · Reviewer_xTGM · 2025-10-30

**Soundness:** 2
**Presentation:** 2
**Contribution:** 2
**Rating:** 2
**Confidence:** 4

**Summary:**

The paper extends the use of sparse autoencoders (SAEs) beyond static interpretability by tracking how SAE-extracted features evolve during the training of Transformer models. Since some models release intermediate checkpoints, the authors propose training a sequence of SAEs - one per checkpoint - each initialized from the previous one to reduce training cost, and they call this method track-SAE. Using this framework, they analyze how features change both semantically and geometrically over time. They distinguish, via human annotation, token-level and concept-level features, and identify three phases of evolution (warmup, emergent, convergent), as well as transformation patterns (maintaining, shifting, and grouping). They also study how the SAE dictionary evolves across checkpoints.

**Strengths:**

- The track-SAE method gives a way to track features dynamically, assuming that dense checkpoints are available.
- The geometric perspective in fig. 3 is interesting.
- The writing of the paper is well organised, and the limitation section is well documented and clear.

**Weaknesses:**

- A key limitation of the proposed method is its reliance on the assumption that representations do not vary significantly from checkpoint to checkpoint. This assumption only holds for models with very dense checkpoints—a scenario that is often impractical or unavailable for very large models. Consequently, the SAE-track method is applicable to only a narrow subset of models. This not only increases training costs but also limits the analysis of neuron tracking: it would be impossible to track a given feature if a SAE were initialized from scratch at every checkpoint.
- Another limitation concerns the analysis itself: checkpoint indices are used instead of actual training steps. Since your checkpoint indices are spaced unevenly and far apart ( [0, 1, 2, 4, 8, 16, 32, 64, 128, 256, 512] + list(range(1000, 143000 + 1, 1000), line 992), this conflicts with your key intuition (Sec. 2.2) and makes the provided visualizations difficult to interpret. For instance, I question the validity of Conclusion 3.1, which claims that the similarity to the final dictionary evolves in three distinct phases. Plotting checkpoint indices instead of actual training steps could easily distort this perceived behavior.
- The concept of a “Feature Region” is confusing. Why define this quantity if it is not used? Similarly, why include a representation of it in Figure 3? It is unclear why such machinery is necessary to extract the top-k datapoints that align with a particular feature.
- The progress measure is introduced to make conclusions quantitative, yet the results remain largely qualitative, as the measure is tracked for only around 20 features chosen arbitrarily.

**Questions:**

I would like to ask the authors to clarify my two main concerns:

1. Use of Checkpoint Indices vs. Training Steps:  Why were checkpoint indices used instead of actual training steps, and how might this choice affect the interpretation of visualizations and conclusions?

2. Feasibility of Dense Checkpoints: The method assumes very dense checkpoints, which may not be practical or available for reasonably large models. Could the authors clarify how the SAE-track method would hold for the Pythia model analyzed here, given the sparsity of its checkpoints (line 992)?

3. Smaller models: Why didn’t you try to use track-SAE on a shallow transformer ?

---

### Official Review · Reviewer_4dRg · 2025-10-31

**Soundness:** 3
**Presentation:** 3
**Contribution:** 3
**Rating:** 6
**Confidence:** 4

**Summary:**

The authors study the evolution of features during LLM training using Sparse Autoencoders (SAEs). They propose training a continual series of SAEs on the sequential training checkpoints of a pretrained LLM, where the SAE trained on the next checkpoint is initialized with the trained SAE from the previous checkpoint. They then qualitatively and quantitatively study the evolution of features in LLMs, beginning with designing a taxonomy for token- and concept-level features. They introduce a progress measure for feature formation in both activation space and feature space and plot this measure for a variety of features over the course of training, where they highlight three main phases during LLM feature evolution. Finally, they study the feature vectors within the latent space by first analyzing how much they drift during training by comparison with the final feature direction. They then look at the full trajectory during training (when projected onto 2 principal components), again highlighting the phases in feature evolution and also showing that fully "formed" features still continue to drift.

**Strengths:**

The authors present a well-written and comprehensive study of feature evolution during LLM training. They introduce a novel method for training sequences of SAEs on training checkpoints and study the evolution of features during training in a variety of ways. They study both token- and concept-level feature formation qualitatively and quantitatively and provide original analysis of feature trajectories in the latent space. During their analysis, they also propose a new metric to measure the progress of feature formation.

**Weaknesses:**

It is unclear to what degree these results are influenced by the continual training approach used for the SAEs. There should be a comparison with training SAEs separately on each checkpoint to determine a) how similar the features from those SAEs are to those from the continually trained models and b) how the feature evolution looks for the standard approach. Additionally, quality measures should be reported to evaluate how well the SAEs are capturing the LLMs' representations.

Some analysis is a little unclear and needs clarification, particularly when studying the three phases in feature evolution.

**Questions:**

### General

Why did you choose to focus on layer 4?

### Continual Training

Can you report metrics quantifying SAE quality over the course of this continual setup? It's unclear how good the different SAEs are at different checkpoints and how they compare to one another.

### Progress Measures

Does the feature space progress measure use cosine similarity?

It appears that all of the activation space measures end at a middle level of similarity and don’t converge to a level much higher than how they started. Why is that? Did you investigate this further?

It also does not seem like the trajectories are particularly stable at the end in the activation space. There do appear to be three phases, but I don’t see any I would describe as “low-rise-stable” in the activation space. What do you mean when you describe the final portion as stable?

How did you choose the annotated features to highlight in the top of Figure 4?

### Trajectory Analysis

What is the quantitative threshold for a feature to be considered "formed" or in the "maintaining" phase?

Some features seem to initialize already formed, why is that?

---

### Official Review · Reviewer_tyEm · 2025-11-01

**Soundness:** 2
**Presentation:** 3
**Contribution:** 2
**Rating:** 4
**Confidence:** 2

**Summary:**

- Proposes SAE-Track, a novel framework for tracking the evolution of sparse autoencoder (SAE) features throughout LLM training.

- Identifies three training phases (Initialization, Emergent, Convergent) and three transformation patterns (Maintaining, Shifting, Grouping) in the semantic evolution of features.

- Provides quantitative analyses on feature formation and feature drift using open-source models (Pythia, GPT-2).

- Aims to establish a unified, continuous framework for mechanistic interpretability beyond static snapshots.

**Strengths:**

- Addresses an important interpretability gap by enabling continuous tracking of feature evolution rather than static analyses.

- Combines semantic, geometric, and dynamical perspectives, offering a comprehensive view of feature development.

**Weaknesses:**

- The experiments rely on relatively small open-source models, which may limit the generalizability of the conclusions to larger LLMs.

- The definition and identification of the three phases could be clarified. It may be useful to compare how the phases emerge across different perspectives (semantic concepts, feature dynamics, decoder vector evolution) and whether they match up to ensure consistency.

- Although some results for additional layers are provided in the appendix, it would strengthen the work to include a direct comparison across layers; do similar trends hold across all layers, or are there systematic differences?

- It would be helpful to include a discussion on the practical utility of tracking feature evolution or the observed results, e.g., how these insights could inform training optimization or interpretability interventions.

**Questions:**

- Could the analysis be extended to better support layer-wise comparisons?

- In Figure 4, how were the plotted features selected, and could you provide more details on how the annotations were obtained?

---

### Meta-Review · Area_Chair_h1xj · 2026-01-06

**Summary:**

In the absence of a rebuttal, most of reviewer scores remain negative, and the concerns raised during the initial review period remain unresolved.

**Reviewer Concerns:**

In the absence of a rebuttal, most of reviewer scores remain negative, and the concerns raised during the initial review period remain unresolved.

**Reviewer Scores:**

In the absence of a rebuttal, most of reviewer scores remain negative, and the concerns raised during the initial review period remain unresolved.

---

### Decision · Program_Chairs · 2026-01-26

Reject